

# 1 Sources and oxidative potential of water-soluble humic-like substances
# 2 (HULIS$_{WS}$) in fine particulate matter (PM$_{2.5}$) in Beijing

Yiqiu Ma[1,2], Yubo Cheng[2], Xinghua Qiu*[,1], Gang Cao[3], Yanhua Fang[1], Junxia Wang[1], Tong Zhu[1], Jianzhen
Yu[4], Di Hu*[,2,5,6]
[1]State Key Joint Laboratory for Environmental Simulation and Pollution Control, College of Environmental Sciences and
Engineering, Peking University, Beijing 100871, P. R. China
[2]Department of Chemistry, Hong Kong Baptist University, Kowloon Tong, Kowloon, Hong Kong, P. R. China
[3]Harbin Institute of Technology (Shenzhen), Shenzhen Key Laboratory of Organic Pollution Prevention and Control, Shenzhen
518055, P. R. China
[4]Department of Chemistry, Hong Kong University of Science and Technology, Clear Water Bay, Kowloon, Hong Kong, P. R.
China
[5]State Key Laboratory of Environmental and Biological Analysis, Hong Kong Baptist University, Kowloon Tong, Kowloon, Hong
Kong, P. R. China
[6]HKBU Institute of Research and Continuing Education, Shenzhen Virtual University Park, Shenzhen, 518057, P. R. China
*Correspondence to*: Xinghua Qiu (xhqiu@pku.edu.cn); Di Hu (dihu@hkbu.edu.hk)
**Abstract.** Water-soluble humic-like substances (HULIS$_{WS}$) are a major redox-active component of ambient fine particulate matter
(PM$_{2.5}$); however, information on their sources and associated redox activity is limited. In this study, total HULIS$_{WS}$, various
HULIS$_{WS}$ species, and HULIS$_{WS}$-associated dithiothreitol (DTT) activity were quantified in PM$_{2.5}$ samples collected during a 1-
year period in Beijing. Strong correlation was observed between HULIS$_{WS}$ and DTT activity; both exhibited higher levels during
the heating season than during the non-heating season. Positive matrix factorization analysis of both HULIS$_{WS}$ and DTT activity
was performed. Four combustion-related sources, namely coal combustion, biomass burning, waste incineration, and vehicle
exhaust, and one secondary factor were resolved. In particular, waste incineration was identified as a source of HULIS$_{WS}$ for the
first time. Biomass burning and secondary aerosol formation were the major contributors (>59%) to both HULIS$_{WS}$ and associated
DTT activity throughout the year. During the non-heating season, secondary aerosol formation was the most important source,
whereas during the heating season, the predominant contributor was biomass burning. The four combustion-related sources
accounted for >70% of HULIS$_{WS}$ and DTT activity, implying that future reduction in PM$_{2.5}$ emissions from combustion activities
can substantially reduce the HULIS$_{WS}$ burden and their potential health impact in Beijing.

## 28 1 Introduction

Air pollution caused by ambient fine particulate matter (PM$_{2.5}$) is a significant environmental problem worldwide. PM$_{2.5}$ that carries
various pollutants may be transported into the human respiratory and thus lead to myriad health effects (Becker et al., 2005; Nel,
2005). Mechanism of these health effects isn't fully understood up to date; however, perturbing the redox equilibrium through the
generation of excessive reactive oxygen species (ROS) is considered as a fundamental way, which has been proposed to be related
with the high redox-active components in PM$_{2.5}$. Transition metals and quinones are two such components with high oxidative
potential (Charrier and Anastasio, 2012; Chung et al., 2006). More recently, another abundant water-soluble organic component,
i.e. humic-like substances (HULIS$_{WS}$), have also been recognized to be highly redox-active (Dou et al., 2015; Lin and Yu, 2011;
Verma et al., 2015a).
HULIS$_{WS}$ are a mixture of compounds that contain polycyclic ring structures with aliphatic side chains and multiple functional
groups, and a significant proportion (30%–80%) of the water-soluble organic matter (WSOM) in PM$_{2.5}$ (Graber and Rudich, 2006;



Kuang et al., 2015; Lin et al., 2010a). The reversible redox sites in HULIS$_{WS}$ fraction could serve as electron transfer intermediary
and lead to continuous production of ROS (Lin and Yu, 2011). Actually, many recent studies have reported the significant role of
HULIS$_{WS}$ in driving PM-associated ROS formation (Dou et al., 2015; Lin and Yu, 2011; Verma et al., 2015a). Dithiothreitol (DTT)
assay is frequently applied to evaluate the oxidative potential of PM$_{2.5}$ components, especially for organic compounds (Xiong et
al., 2017). By adopting this method, Verma et al. (2015b) found that HULIS$_{WS}$ caused approximately 45% of DTT activity of the
water extract from PM$_{2.5}$ sampled in Atlanta, USA. This was 5% higher than that induced by water-soluble metals (Verma et al.,
2015b). Furthermore, the DTT activity of HULIS$_{WS}$ is about 79% of the whole WSOM fraction in PM$_{2.5}$ samples (Lin and Yu,
2011), suggesting a substantial health threat induced by HULIS$_{WS}$. Thus, given the considerable amount of HULIS$_{WS}$ in PM$_{2.5}$ and
their high ROS generation ability, both field measurements and smog chamber experiments have been conducted to determine
their formation mechanisms and atmospheric origins (Kautzman et al., 2010; Lin et al., 2010b; Sato et al., 2012); biomass burning
and secondary formation have been suggested to be the major sources (Kautzman et al., 2010; Lin et al., 2010b). However, until
now, studies on the quantitative source apportionment of HULIS$_{WS}$ remain relatively rare (Kuang et al., 2015),  and information
on the source-specific contribution to their redox activity is lacking.
Beijing, the capital of China located in the North China Plain, is a political and cultural center with densely population. On the
other hand, it has become one of the most polluted cities in the world, with an annual PM$_{2.5}$ concentration of up to 89.5 μg m$^{-3}$ in
2013 (Li et al., 2017). Therefore, it presents an ideal location to study the chemical characteristics of HULIS$_{WS}$ as well as their
sources and potential redox activity.
In this study, our major objective is to investigate the ROS-forming ability of HULIS$_{WS}$ in relation to different sources and
meteorological conditions. Thus, a total of 66 PM$_{2.5}$ samples collected in Beijing during a 1-year period were analyzed.
Concentrations of total HULIS$_{WS}$ were quantified, together with some characteristic individual HULIS$_{WS}$ species and the major
aerosol components. HULIS$_{WS}$-associated redox activity was also determined using a DTT assay. Positive matrix factorization
(PMF) analysis was applied to determine the sources of both HULIS$_{WS}$ and their associated redox activity. Such a comprehensive
source apportionment study of HULIS$_{WS}$-related ROS-generation potential has not been previously reported. Results from this
study could provide not only quantitative information regarding the sources and toxicity of HULIS$_{WS}$, but also a deeper
understanding of the source-specific oxidative potential of Chinese urban organic aerosols in general. This may be useful for the
future development of source-targeted air pollution control policies in Beijing and may provide public-health benefits.

## 65 2 Material and methods

### 66 2.1 Sample collection

PM$_{2.5}$ samples were collected at the Peking University Atmosphere Environment Monitoring Station (PKUERS) on the campus of
Peking University (39°59'21"N, 116°18'25"E, approximately 30 m above the ground), Beijing, China. A high-volume air sampler
coupled with a ≤2.5 μm inlet (HIVOL-CABLD, ThermoFisher Scientific, Waltham, MA, USA) was used to conduct sampling at
a flow rate of 1.13 m$^3$ min$^{-1}$. Samples were collected on quartz fiber filters (20.3 × 25.4 cm$^2$, prebaked at 550 °C for 5 h; Whatman,
Hillsboro, OR, USA) for 24 h every 6 days from 3 March 2012 to 1 March 2013. In addition, a four-channel mid-volume sampler
was operated synchronously (16.7 L min$^{-1}$, TH-16A, Wuhan Tianhong Instruments Co. Ltd, China) to collect PM$_{2.5}$ onto three 47-
mm Teflon filters and one quartz fiber filter for the determination of PM$_{2.5}$ mass, elemental carbon (EC) and organic carbon (OC),
and inorganic ionic species.





## 2.2 Chemical analysis

$HULIS_{WS}$ was isolated from $PM_{2.5}$ samples following the procedure described by Lin et al. (2010b). Briefly, a portion of sample filters (17.5 cm² for individual $HULIS_{WS}$ species and 3 cm² for total $HULIS_{WS}$) was cut into small pieces and pollutants were extracted through sonication with distilled deionized (DDI) water for 30 min. The extracts were filtered with polytetrafluoroethylene (PTFE) filters (0.45-µm pore size; Grace, Houston TX, USA) and acidified to a pH of 2 with 2.4 M HCl. A solid phase extraction (SPE) cartridge (Oasis HLB, 3 mL/30 µm, 60 mg; Waters, Milford, MA, USA) was used to isolate $HUILS_{WS}$. The SPE cartridge was first activated using 1.0 mL of methanol and equilibrated using 1.0 mL of 0.01 M HCl. The extracts were then loaded onto an HLB cartridge. Because the majority of inorganic ions, low molecular weight organic acids, and sugar compounds could not be retained by the HLB cartridge, they were removed from the final effluent. For the analysis of individual $HULIS_{WS}$ species, the HLB cartridge was rinsed with two 1.0-mL portions of DDI water and then eluted with three 0.5-mL portions of basic methanol (2% ammonia, w/w). The effluent was dried with a gentle flow of ultrapure nitrogen at 40 °C, and then derivatized with 100 µL of N,O-bis(trimethylsilyl)trifluoroacetamide (BSTFA; with 1% trimethylchlorosilane; Sigma Aldrich, St. Louis, MO, USA) and 50 µL of pyridine (>99.5%; International Laboratory USA, CA, USA) at 70 °C for 2 h. When the mixture had cooled to room temperature, it was spiked with 30 µL of tetracosane-d₅₀ (50 µg mL⁻¹ in n-hexane; Sigma Aldrich, St. Louis, MO, USA) as the internal standard for gas chromatography-mass spectroscopy (GC-MS; 5975-7890A, Agilent, Santa Clara, CA, USA) analysis. Detailed information on this analysis is provided in the Supplementary Material.

For the analysis of total $HULIS_{WS}$, 1.5 mL of basic methanol (2% ammonia, w/w) was replaced by 6.0 mL of pure methanol in SPE step to avoid possible influence of ammonia in the following DTT experiments (Lin and Yu, 2011). The larger amount of solvent was to maintain the elution efficiency (Lin and Yu, 2011). Detailed information for the GC-MS quantification intensity of individual $HULS_{WS}$ species eluted by the two protocol was provided in the Supplementary Material. The effluent was dried with nitrogen, and restored in 1 mL of DDI water for quantification. An aliquot of 20 µL of aqueous solution was injected into a high-performance liquid chromatography system (HPLC, ThermoFisher Scientific, Waltham, MA, USA) coupled with an evaporative light scattering detector (Alltech ELSD 3300, Grace, Houston, TX, USA). Since ELSD is mass sensitive, the mass of $HULIS_{WS}$ instead of $HULIS_{WS\_carbon}$ was reported in this study. Detailed information on the HPLC-ELSD conditions is provided in the Supplementary Material.

Major water-soluble ions were identified and quantified using ion-chromatography (DIONEX, ICS-2500 for cations and ICS-2000 for anions, ThermoFisher Scientific, Waltham, MA, USA, Tang et al., 2011). EC and OC were analyzed by a thermal–optical carbon analyzer (Sunset Laboratory-Based Instrument, Tigard, OR, USA, Tang et al., 2011). Hopanes were measured by in-injection thermal desorption-gas chromatography mass spectrometry (GC-MS, Agilent 6890N-5975C, Santa Clara, CA, USA, Ho and Yu, 2004), while levoglucosan was measured using an Agilent 7890A- 5975C GC-MS (Hu et al., 2008).

## 2.3 DTT assay

The procedure of the DTT assay follows that used by Li et al. (2009) and Lin and Yu (2011). A 120-µL portion of $HULIS_{WS}$ solution was transferred into an eppendorf tube. Then 920 µL of potassium phosphate buffer (pH = 7.4) containing 1 mM diethylene triamine pentaacetic acid (DTPA) and 50 µL of 0.5 mM DTT (both >99%; Sigma Aldrich, St. Louis, MO, USA) were added and mixed thoroughly. The samples were subsequently placed in a dry bath at 37 °C for 90 min and spiked with 100 µL of 1.0 mM 5,5'-dithiobis-2-nitrobenzoic acid (DTNB, 98%; Sigma Aldrich, USA) containing 1 mM DTPA. Finally, the absorbances of the reacted sample solutions were measured at 412 nm within 30 min using an ultraviolet-visible (UV-Vis) spectrophotometer (8453, Hewlett Pakard, Palo Alto, CA, USA). Considering that some transition metals may still remained in the $HULIS_{WS}$ fraction even after HLB purification, sufficient amount of DTPA was added in the procedure to chelate all the remaining transition metals, such



as Cu, Mn, and Fe, to eliminate the redox-activity induced by these metals (Lin and Yu, 2011). For the control samples, blank
filters were used instead of real samples.
Based on previous experiments, the time-dependent consumption of DTT catalysed by $HULIS_{WS}$ is linear when DTT consumption
is less than 90% (Lin and Yu, 2011). In this study, verification experiments were also performed with a similar result. Give that
the DTT consumption rates for all the 66 samples in this study were between 3.6% and 77.0%, the DTT activity is proportionally
related to $HULIS_{WS}$ concentration.
**2.4 Source apportionment**
In this study, United States Environmental Protection Agency PMF 5.0 was applied to identify and apportion the sources of both
$HULIS_{WS}$ and $HULIS_{WS}$-associated redox activity. As suggested by Henry et al. (1984), the minimum sample size of N for PMF
analysis was $30 + (V + 3)/2$, where V is the number of input species. A total of 66 samples and 13 species were included in PMF
analysis, which was an adequate sample size to obtain a statistically reliable PMF result. Details of PMF parameter settings are
provided in the Supplementary Material.
**3 Results and discussion**
**3.1 Total $HULIS_{WS}$ and $HULIS_{WS}$-associated DTT activity**
In this study, the concentrations of total $HULIS_{WS}$ and $HULIS_{WS}$-associated DTT activity in 66 $PM_{2.5}$ samples were quantified.
The annual average concentration of total $HULIS_{WS}$ in Beijing measured in this study was 5.66 µg m$^{-3}$ (median: 4.30, range: 1.08–
22.36 µg m$^{-3}$). This was approximately 20% higher than those measured in three other Chinese cities: 4.83 µg m$^{-3}$ in Guangzhou
(Kuang et al., 2015), 4.71 µg m$^{-3}$ in Nansha (Kuang et al., 2015), and 4.69 µg m$^{-3}$ in Lanzhou (Tan et al., 2016). A clear temporal
variation of total $HULIS_{WS}$ was observed (Figures 1, 2), with significantly higher levels ($p < 0.05$, Mann–Whitney test) in the
heating season (November through March; average 7.93, median 6.15 µg m$^{-3}$) than in the non-heating season (April through
October; average 3.72, median 2.86 µg m$^{-3}$). This could be mostly attributed to the intensive coal and biomass burning activities
performed for residential heating during the heating season. In addition, lower temperatures and mixing heights during the heating
season could also favor the formation of particle-bound $HULIS_{WS}$ species. However, the contributions of total $HULIS_{WS}$ to organic
matter (OM, calculated by OC multiply the ratio of 1.98 and 1.50 for the heating and non-heating seasons, respectively, Xing et
al., 2013) in $PM_{2.5}$ are slightly lower during the heating season (21.8% ± 13.5%) than that during the non-heating season (27.4% ±
12.0%, Figure 1), indicating higher levels of other combustion-generated organic compounds were emitted in the heating seasons
other than $HULIS_{WS}$.
For $HULIS_{WS}$-associated DTT activity, they exhibited similar temporal variation as $HULIS_{WS}$ (Figure 2), with significantly higher
levels in the heating season (average 0.073, median 0.063 nmol min$^{-1}$ m$^{-3}$) than in the non-heating season (average 0.031, median
0.029 nmol min$^{-1}$ m$^{-3}$). Because most of the inorganic ions were not retained by the HLB cartridge and the remaining metals in
the $HULIS_{WS}$ effluent were chelated by DTPA, the DTT activity measured here could be attributed entirely to $HULIS_{WS}$. In fact,
a strong correlation between total $HULIS_{WS}$ and $HULIS_{WS}$-associated DTT activity was observed ($R^2 = 0.78$).
**3.2 Individual species of $HULIS_{WS}$**
Because the main objective of this study was to identify the sources of $HULIS_{WS}$ and their associated redox activity, we mainly
focused on the identification of organic markers in the chemical analysis. A total of 25 species were identified and quantified in
the $HULIS_{WS}$ fraction of $PM_{2.5}$ through GC-MS, including 12 aromatic acids, five nitrophenol analogues, three aliphatic acids, and
five biogenic secondary organic aerosol (SOA) tracers (Table S1 in the Supplementary Material, Hu et al., 2008)





All 12 aromatic acids, including three hydroxyl benzoic acids, three benzenedicarboxylic acids, three benzenetricarboxylic acids,
2-hydroxy-5-nitrobenzoic acid, vanillic acid, and syringic acid, exhibited higher levels during the heating season than during the
non-heating season (Figure S2 in the Supplementary Material). Among these acids, terephthalic acid (TPha) was the most abundant
(average 150.2 ng m$^{-3}$ in the heating season, and 98.1 ng m$^{-3}$ in the non-heating season), accounting for approximately 2% of the
total HULIS$_{WS}$. Compared with other Chinese cities, the concentration of TPha in Beijing was substantially higher than those in
the southern cities such as Hong Kong (19.9 ng m$^{-3}$ in winter, Ho et al., 2011) and similar to those in the northern cities such as
Xi'an (54 ng m$^{-3}$ in summer and 250 ng m$^{-3}$ in winter, Cheng et al., 2013). TPha is mainly used to produce
polyethyleneterephthalate (PET) plastics, which are widely used for bottles and containers; therefore, it has been suggested as a
tracer for the pyrolysis of domestic waste (Kawamura and Pavuluri, 2010; Simoneit et al., 2005). Meanwhile, benzenetricarboxylic
acids were considered to be secondarily formed from the photodegradation of organic precursors such as polycyclic aromatic
hydrocarbons (PAHs) (Kautzman et al., 2010). Therefore, 1,2,3-benzenetricarboxylic acid (123Ben) and 1,2,4-
benzenetricarboxylic acid (124Ben) were also included in the PMF analysis.
Similar to the aromatic acids, all five nitrophenol analogues, namely 4-nitrophenol, 2-nitrocatechol, 2-methyl-4-nitrophenol
(2M4NP), 4-methly-5-nitrocatechol (4M5NC), and 3-methly-6-nitrocatechol (3M6NC), exhibited 8–14 times higher
concentrations during the heating season than during the non-heating season (Table S1 in the Supplementary Material). In
particular, 4M5NC and 3M6NC not only showed similar temporal variations but also were strongly correlated (R$^2$ = 0.87), implying
that they may have similar sources. These two compounds have been suggested as tracers for the aging process of biomass burning
(Iinuma et al., 2010; Kahnt et al., 2013). However, Iinuma et al. (2010) pointed out that the photo-oxidation of vehicle exhaust
may be a more significant source for these two compounds in urban areas. Given that both 4M5NC and 3M6NC are good
anthropogenic SOA markers, they were also included in the PMF analysis.
Five biogenic SOA tracers including 3-hydroxyglutaric acid, 3-hydroxy-4,4-dimethylglutaric acid, 3-methyl-1,2,3-
butanetricarboxylic acid, 3-isopropylglutaric acid, and 3-acetylglutaric acid were identified and quantified. Because they were all
formed from the atmospheric oxidation of monoterpenes and had similar temporal variations, they were grouped as SOA markers
of monoterpene (MonoT) in the PMF analysis (Hu et al., 2010). Briefly, MonoT showed higher concentrations during the non-
heating season (average 16.9, median 15.2 ng m$^{-3}$) than during the heating season (average 12.5, median 10.2 ng m$^{-3}$), which was
opposite to that of total HULIS$_{WS}$. Because of the higher biogenic volatile organic compounds (VOCs) emissions, more intense
solar radiation, and higher temperature and humidity in the non-heating season, more active secondary formation could lead to
higher concentrations of biogenic SOA (Guo et al., 2012).
**3.3 Source apportionment of total HULIS$_{WS}$ and their ROS activity**
The optimal PMF solution was determined with five factors (A–E; Figure 3). The Q$_{robust}$ obtained was 62.9, which was exactly
equal to Q$_{true}$, and the scaled residues for all species were between −2 and +2, indicating no outliers for this solution. Constrained
model operation was adopted for a more reasonable interpretation (dQ$_{robust}$% = 0.32%) (Norris et al., 2014). The optimized solution
was bootstrapped 100 times, with 100% of the runs producing the same factors. A strong linear correlation between the measured
and PMF-predicted HULIS$_{WS}$ concentrations (R$^2$ = 0.76) also suggested a reliable PMF solution (Figure S4 in the Supplementary
Material).
As shown in Figure 3, factor A had a high percentage of non-sea salt Cl$^-$ (nss-Cl$^-$, [nss-Cl$^-$] = [Cl$^-$] − 1.17 × [Na$^+$]), and was
attributed to coal combustion (Tan et al., 2016; Tao et al., 2016; Zhang et al., 2013). Factor B had a high loading of levoglucosan
and was determined as biomass burning (Hu et al., 2010; Tao et al., 2016). Factor C was considered to be waste incineration, due
to the high level of TPha. Factor D was dominated by hopanes, tracers for fuel combustion, suggesting traffic related activities (Hu





et al., 2010). In particular, the two anthropogenic markers, 4M5NC and 3M6NC, were mostly assigned to this factor (4M5NC
46%, and 3M6NC 33%) instead of factor C (4M5NC 14%, and 3M6NC 25%). These two species were mainly formed through the
photo-oxidation of cresols, which were directly emitted through wood combustion or produced from toluene through its reaction
with OH radicals in the presence of $NO_X$ (Iinuma et al., 2010). Traffic emissions were a significant source for single-ring aromatics,
especially toluene, in Chinese megacities (Huang et al., 2015). In this study, the sampling site was located in an urban area
influenced by considerable vehicular emissions of $NO_X$ and toluene, which may have led to subsequent secondary formation of
4M5NC and 3M6NC. Therefore, the fourth factor was considered as a mixed source including both primary emission and the aging
process of traffic exhaust. The fifth factor was characterized by a predominant loading of MonoT, $SO_4^{2-}$, and $NH_4^+$; thus, it was
considered as a secondary aerosol formation source.

**3.4 Source-specific contributions to HULIS$_{WS}$**

Source-specific contributions to HULIS$_{WS}$ during both non-heating and heating seasons were calculated based on PMF results.
The four combustion-related sources contributed >80% of HULIS$_{WS}$ in the heating season and 50% in the non-heating season
(Figure 4A), of which biomass burning was the most predominant. A strong correlation ($R^2$ = 0.51, Figure S5 in the Supplementary
Material) was observed between HULIS$_{WS}$ and levoglucosan, a marker of biomass burning, and this was consistent with previous
studies (Lin et al., 2010b). Approximately 33% of total HULIS$_{WS}$ was attributed to biomass burning during the 1-year sampling
period in Beijing, higher than that observed in the Pearl River Delta region (8%−28%, Kuang et al., 2015). The intensive wood
and crop residue burning activities in the Beijing−Tianjin−Hebei region during autumn and winter could emit a large amount of
aerosols into the atmosphere (Zhang et al., 2013). Thus, as shown in Figure 4A, the contribution of biomass burning to HULIS$_{WS}$
in the heating season (2.96 μg m$^{-3}$) was 3.5 times that in the non-heating season (0.84 μg m$^{-3}$).
A previous study reported that refuse burning may contribute 1%–24% of organic particles in Asia (Simoneit et al., 2004). In this
study, waste incineration was found for the first time as an important source of HULIS$_{WS}$ in Beijing, with a considerable and stable
contribution to total HULIS$_{WS}$ throughout the year (18.7% in the non-heating season and 17.1% in the heating season). According
to the China Statistic Yearbook (2013), 6.33 million tons of domestic waste were produced in Beijing during 2012 (National Bureau
of Statistics of China, 2013), among which 0.95 million tons were disposed of through incineration. Given that nearly 24% of the
urban waste was plastic (Wang and Wang, 2013), the incineration of such large amounts of domestic waste may explain the high
levels of TPha and other HULIS$_{WS}$ compounds in Beijing.
Coal has occupied the predominant position in China's energy consumption for a long time (Zhang and Yang, 2013). Therefore,
coal combustion is an important source of PM$_{2.5}$ pollution in China, especially in northern Chinese cities. Tan et al. (2016) identified
a strong correlation between HULIS$_{WS}$ and Cl$^-$ ($R^2$ = 0.89) in Lanzhou and suggested that coal burning was probably the major
contributor to HULIS$_{WS}$ in winter. However, the contribution of coal combustion to HULIS$_{WS}$ was found to be minor (5.8%) in
the present study. Similarly, a source apportionment analysis of PM$_{2.5}$-bound water-soluble organic carbon (WSOC) in Beijing
found that less than 5% of WSOC was from coal combustion (Tao et al., 2016). This was because less oxidized compounds
including polycyclic aromatic compounds were favorably produced from the aromatic fragments of coal under the fuel-rich
incomplete combustion conditions; these less oxidized compounds are generally hydrophobic substances and not extracted into the
HULIS$_{WS}$ fraction.
A correlation between total HULIS$_{WS}$ and hopanes ($R^2$ = 0.46, Figure S5 in the Supplementary Material) might suggest direct
emissions of HULIS$_{WS}$ from vehicle exhaust. As shown in Figure 4A, vehicle emissions are responsible for 13.7% of PM$_{2.5}$-bound
HULIS$_{WS}$. Interestingly, the amount of HULIS$_{WS}$ assigned to vehicle exhaust was approximately three times higher in the heating
season than in the non-heating season (Figure 4A). This could be attributed to the low temperature in winter, which favors the



partition of semivolatile HULIS$_{WS}$ species into particle phases. Another explanation could be that more HULIS$_{WS}$ were formed
from the aging process of traffic exhaust in the heating season. To evaluate this hypothesis, multilinear regression (MLR) analysis
was conducted to assess the effects of NO$_X$, O$_3$, SO$_4^{2-}$, particle acidity (H$_p^+$), and particle-phase liquid water content (LWC$_p$) on
the HULIS$_{WS}$ resolved in the vehicle emissions factor (HULIS$_{WS\_VE}$; the calculation of H$_p^+$ and LWC$_p$, and the MLR analysis
results are provided in the Supplementary Material). NO$_X$ was found as the only statistically significant factor that was positively
correlated to HULIS$_{WS\_VE}$ with a regression coefficient of 0.012 (p < 0.001; Table S2 in the Supplementary Material), suggesting
that a 1 μg m$^{-3}$ increase in NO$_X$ was associated with a 0.012 μg m$^{-3}$ increase in HULIS$_{WS\_VE}$, when holding other covariates
unchanged. In fact, vehicle exhaust was the major source of ground level NO$_X$ (>60%) in Beijing, even in the heating season (Lin
et al., 2011). A higher level of NO$_X$ was observed during the heating season than during the non-heating season due to a lower
boundary layer and weaker vertical mixing (Figure S6 in the Supplementary Material). Kautzman et al. (2010) found that ring-
opening oxygenated products with one benzyl group, which could be retained by the HLB cartridge and were considered as
HULIS$_{WS}$ components, were predominantly formed from the photo-oxidation of PAHs under high NO$_X$ conditions. Thus, the
higher levels of NO$_X$ in the heating season led to higher levels of secondarily produced HULIS$_{WS\_VE}$, indicating a synergistic effect
of primary emission and the secondary aging process from vehicle exhaust. Furthermore, the presence of 4M5NC and 3M6NC,
SOA markers of cresol, in this factor confirmed that a certain fraction of HULIS$_{WS\_VE}$ was secondarily formed.
In addition to the four combustion-related sources, one secondary source was apportioned by PMF, contributing 30.1% of
HULIS$_{WS}$ throughout the year. MLR analysis was conducted to evaluate the effects of O$_3$, NO$_X$, SO$_4^{2-}$, H$_p^+$, and LWC$_p$ on the
secondary formation of HULIS$_{WS}$ (HULIS$_{WS-SEC}$). Sulfate was found to be the most significant factor with a regression coefficient
of 0.066 (Table S3 in the Supplementary Material). This may be due to the predominant role of sulfate in the particle-phase
formation of organosulfates, one important HULIS$_{WS}$ component (Xu et al., 2015), through both nucleophilic addition reactions
and the salting-in effect (Lin et al., 2012; Riva et al., 2015). Results from the MLR analysis also indicated that an increase of 1 μg
m$^{-3}$ O$_3$ led to an increase of 0.028 μg m$^{-3}$ HULIS$_{WS\_SEC}$. Gaseous highly oxidized multifunctional organic compounds (HOMs)
were characterized in the ozonolysis of α-pinene in smog chamber experiments (Zhang et al., 2015). It was suggested that, after
partitioning to the particle phase, these HOMs could undergo rapid accretion reactions to form oligomers containing multiple
carboxylic acid and ester groups, which served as good HULIS$_{WS}$ candidates. Considering the higher concentrations of O$_3$ in the
non-heating season (Figure S7 in the Supplementary Material), together with higher biogenic VOCs emissions and temperature as
well as more intense solar radiation, a larger amount of HULIS$_{WS\_SEC}$ was produced in the non-heating season (2.01 μg m$^{-3}$) than
in the heating season (1.41 μg m$^{-3}$).
**3.5 Source-specific contributions to DTT activity**
To gain quantitative insights into the potential health impacts of different HULIS$_{WS}$ sources, source-specific contributions to
HULIS$_{WS}$-associated DTT activity were assessed using PMF result. The strong correlation (R$^2$ = 0.78; Figure S4 in the
Supplementary Material) between measured and predicted DTT activity suggested reliable predictions.
Similar to the source apportionment results of HULIS$_{WS}$, biomass burning was identified as the major contributor to HULIS$_{WS}$-
associated DTT activity in the heating season, and secondary formation was the most important source in the non-heating season
(Figure 4B). The four combustion-related sources accounted for 75% of HULIS$_{WS}$-associated redox activity throughout the year,
of which biomass burning contributed 33.6%, followed by vehicle emissions (18.5%), waste incineration (18.5%), and coal
combustion (4.1%). During biomass burning, highly oxidized organic compounds with quinone, hydroxyl, and carboxyl groups
were directly produced (Fan et al., 2016). Moreover, some of the VOCs emitted from biomass burning could undergo further
reactions and generate high redox-active products, for example, hydroxyquinones formed through ●OH radical oxidation



(McWhinney et al., 2013). Those compounds, such as quinones and hydroxyquinones, could be extracted in HULIS$_{WS}$ fraction and
lead to DTT consumption (Chung et al., 2006; Verma et al., 2015a). Additionally, as reported by Dou et al. (2015) the nitrogen-
containing alkaloids emitted from biomass burning could also enhance the ROS-generation ability of HULIS$_{WS}$.
To further investigate the intrinsic ROS-generation ability of HULIS$_{WS}$, the DTT consumption rate was normalized for HULIS$_{WS}$
mass (DTT$_m$, expressed in units of pmol min$^{-1}$ per µg HULIS$_{WS}$ (Verma et al. 2014). The average intrinsic DTT activity of
HULIS$_{WS}$ in Beijing was 9.91 pmol min$^{-1}$ per µg HULIS$_{WS}$ (median 9.02, range 2.74–25.8 pmol min$^{-1}$ per µg HULIS$_{WS}$), which
was higher than the reported average DTT$_m$ activity (6.4 ± 1.2 pmol min$^{-1}$ per µg HULIS$_{WS}$) of six PM$_{2.5}$ samples collected during
winter in Guangdong, China (Dou et al., 2015). This difference might be attributed to the different chemical components and
sources of HULIS$_{WS}$ in these two regions.
Furthermore, the intrinsic DTT activities of the HULIS$_{WS}$ from the five sources were derived. HULIS$_{WS}$ from vehicle emissions
constituted the most ROS-active HULIS$_{WS}$, with a maximum activity of 12.0 pmol min$^{-1}$ per µg HULIS$_{WS\_VE}$, followed by waste
incineration (9.25 pmol min$^{-1}$ per µg HULIS$_{WS\_WI}$), biomass burning (9.10 pmol min$^{-1}$ per µg HULIS$_{WS\_BB}$), secondary formation
(7.45 pmol min$^{-1}$ per µg HULIS$_{WS\_SEC}$), and coal combustion (6.22 pmol min$^{-1}$ per µg HULIS$_{WS\_CC}$). Similarly, Bates et al. (2015)
revealed that the water-soluble PM$_{2.5}$ (WS-PM$_{2.5}$) from gasoline vehicle emissions had the highest intrinsic DTT activity, probably
due to the oxygenated OC and metals on gasoline particles. Verma et al. (2009) also observed a higher aerosol oxidative potential
from the aged particles of traffic exhaust than those directly emitted, and a strong correlation was observed between oxygenated
organic acids and vehicle-related redox activity. In the present study, vehicle emission was found to be the highest redox-active
source for HULIS$_{WS}$, a large fraction of WS-PM$_{2.5}$. However, because the remaining water-soluble metals in HULIS$_{WS}$ were
chelated through DPTA, the high intrinsic ROS activity of HULIS$_{WS\_VE}$ is believed to be mostly due to the highly oxygenated OC
content in HULIS$_{WS\_VE}$.
Waste incineration was another important primary source of HULIS$_{WS}$-related DTT activity (20.5% in the non-heating season and
17.4% in the heating season), and its intrinsic HULIS$_{WS}$ ROS activity was even slightly higher than that from biomass burning.
Mohr et al. (2009) examined the elemental ratio of aerosols emitted from different sources. They found that particles from plastic
burning had a higher O/C ratio (0.08) than those from diesel (0.03) and gasoline (0.04) combustion, indicating a more oxidized
feature of aerosols emitted through refuse burning (Mohr et al., 2009). Considering that incineration will play an increasingly
important role in waste treatment in Beijing in the following years (National Development and Reform Comission, 2016), concern
should be directed to the potential threat of trash burning to public health.
Although HULIS$_{WS\_SEC}$ was less DTT-active than HULIS$_{WS\_VE}$, HULIS$_{WS\_WI}$, or HULIS$_{WS\_BB}$, secondary aerosol formation served
as the second largest contributor (25.3%) to HULIS$_{WS}$-associated redox activity throughout the year. Higher levels of DTT activity
mediated by HULIS$_{WS\_SEC}$ were observed in the non-heating season (0.015 nmol min$^{-1}$ m$^{-3}$) than in the heating season (0.011 nmol
min$^{-1}$ m$^{-3}$), accounting for 44.1% and 14.5% of HULIS$_{WS}$ DTT activity in each season, respectively. The relatively low intrinsic
DTT activity of HULIS$_{WS\_SEC}$ may be mostly attributed to its abundance of biogenic SOA components such as organosulfates and
organonitrates (Chen et al., 2011), which were found to have negligible ROS-generating ability (Kramer et al., 2016). Although
chamber experiments reported the formation of ROS-active HOMs or organic peroxides through the ozonolysis of biogenic VOCs
(Docherty et al., 2005; Zhang et al., 2015), the production yields of these peroxides were generally low and thus could not have a
major influence on the DTT activity of HULIS$_{WS\_SEC}$.
In summary, four combustion-related sources and one secondary source of PM$_{2.5}$-bound HULIS$_{WS}$ and their associated ROS
potential were identified by PMF in this study. Biomass burning (32.7%) and secondary aerosol formation (30.1%) were the major




contributors to HULIS$_{WS}$ in Beijing. For the first time, waste incineration was identified as an important source of HULIS$_{WS}$, with
a considerable and stable contribution to HULIS$_{WS}$ throughout the year (17.7%). Regarding ROS-generation potential, HULIS$_{WS}$
from vehicle emissions was identified as the most ROS-active, and HULIS$_{WS}$ from secondary aerosol formation showed a lower
intrinsic DTT ability than those of most primary sources except for coal combustion. Such variations in the ROS-generation ability
of HULIS$_{WS}$ from different sources will be relevant for future inquiries into more detailed chemical speciation of HULIS$_{WS}$, their
roles in ROS generation, and the possible oxidation mechanisms involved.
**Supplementary Material.** Information on chemical analysis; PMF source apportionment; MLR analysis together with Table S1-
S3 and Figure S1-S7 are provided.
**Acknowledgement.** This work was supported by the NSFC (21477102 and 41421064), the Joint NSFC-ISF Research Program
(41561144007), the general research Fund of Hong Kong Research Grant Council (12304215, 12300914 and 201212), the Ministry
of Science and Technology of China Grants (973 program; 2015CB553401), the Faculty Research Grant from Hong Kong Baptist
University (FRG2/16-17/041), and Research and Development of Science and Technology in Shenzhen (JCYJ
20140419130357038 and JCYJ 20150625142543472).

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

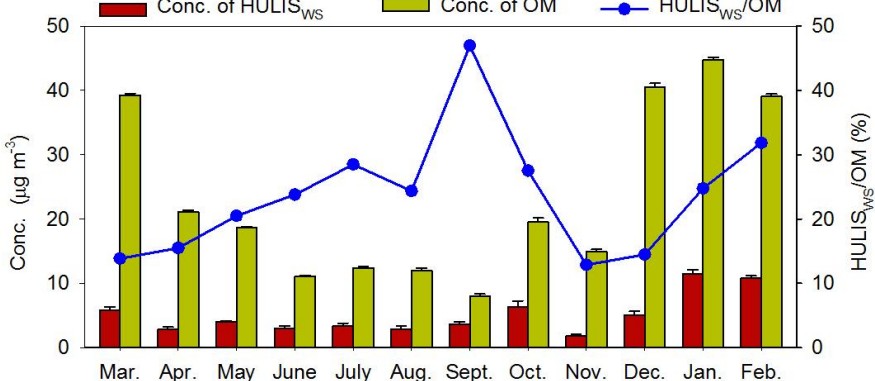

**Figure 1: Monthly average concentrations (average ± standard error) of total HULIS$_{WS}$ and organic matter (OM) in**
**PM$_{2.5}$ collected in Beijing. The monthly percentage contributions of HULIS$_{WS}$ to OM are shown in the blue line.**



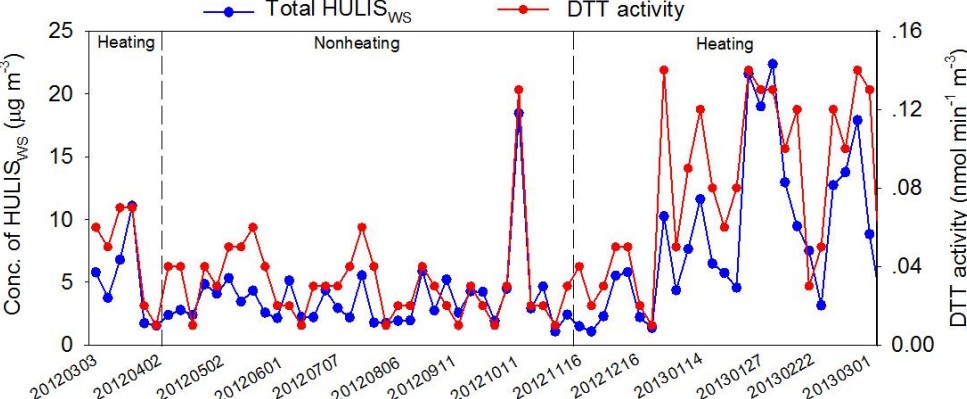


**Figure 2: Temporal variation of total HULIS_WS and associated DTT activity in Beijing.**

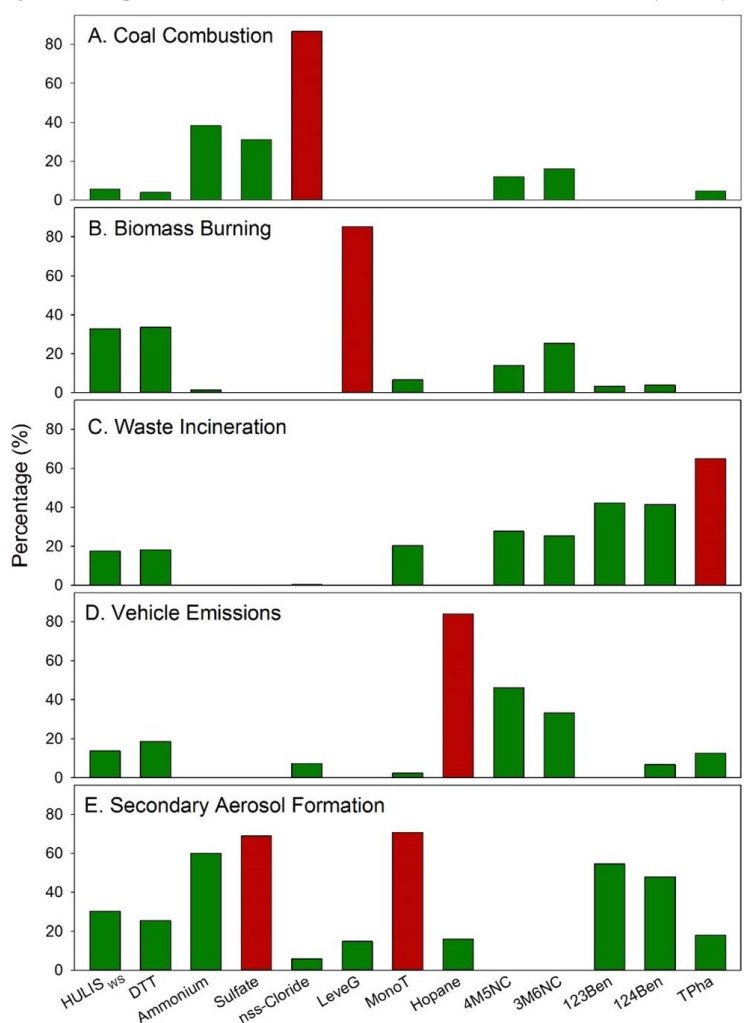


**Figure 3: Distribution of HULIS_WS, HULIS_WS-associated DTT activity and other measured species in the five sources resolved by PMF.**
**Columns in dark red indicate characteristic tracers of each source.**






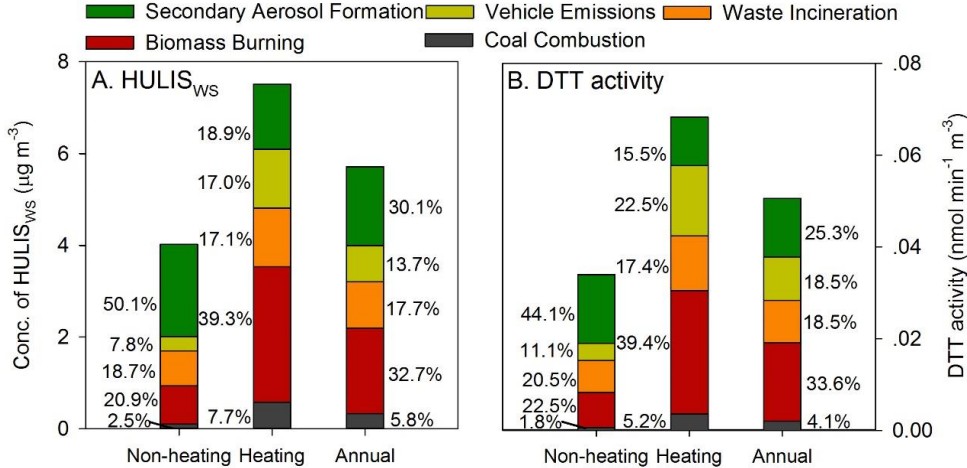


**Figure 4: Source-specific contributions to total HULISWS (panel A) and HULISWS-associated DTT activity (panel B).**
