# Peer review of "Sources and oxidative potential of water-soluble humic-like substances (HULISWS) in fine particulate matter (PM2.5) in Beijing"

_Atmospheric Chemistry and Physics, 2017_

## Referee Comment (RC1) · Anonymous Referee #2 · 5 Dec 2017

It's well known that the toxicity of PM2.5 is greatly related to its chemical composition and pollution sources. This work analyzed PM2.5 samples collected in Beijing during a one-year period, and the levels and oxidative potential (indicated by DTT) of a major water-soluble PM2.5 component, i.e. water-soluble humic-like substances (HULISWS), were reported. With the aid of various characteristic source tracers, PMF was applied to apportion the major sources of both HULISWS and its associated DTT activity. There are some new and interesting findings. The major sources of both HULISWS and DTT activity were coal combustion, biomass burning, traffic exhaust, waste incineration, and secondary formation. Waste incineration was probably identified as a contributor to HULISWS for the first time. Moreover, HULISWS from vehicle emissions was found as
the most ROS-active, and HULISWS from secondary aerosol formation showed a lower intrinsic DTT ability than those of most primary sources except for coal combustion. This paper is well-written. The study is clear, informative, and novel in general, and the major data and their interpretation are scientifically sound. I suggest it to be considered by ACP for publication if the following concerns could be addressed. Line 22: Is it necessary to define the waste as "plastic waste", as terephthalic acid is a marker of plastics? Line 39-40: Add "an" before electron and "a" before continuous. Line 71: If the samples were taken every 6 days for a one-year period, there should be 60 samples. Why there were 66 samples as listed in line 123? Line 100: How many individual hopanes have been identified? I guess the input species "hopane" in PMF must be the sum of all identified hopanes, right? What are the concentrations of hopanes? What are the water-soluble ions idenfied? The author may need to include a table in the supplementary information that provide levels of hopanes, water-soluble ions, EC and OC in the batch of PM2.5 samples analyzed. Line 108: DTPA was spiked to chelate transition metals. Could it also affect or even remove some HULISWS components? Lines 130 to 131: Were all the reference data observed during a one-year period and comparable to present study?

---

## Referee Comment (RC2) · Anonymous Referee #3 · 24 Jan 2018

This paper is of general interest. It provides insights on sources of DTT activity associated with organic species in Beijing. There are, however, some major issues to address. First, the grammar is an issue; the paper needs editing. Secondly, unfortunately the authors apparently quantified DTT activities using a non-standard method, ie that of Li et al., 2009, which differs from the original DTT protocol described by Cho et al. (2005). The Cho method is widely utilized and is the basis for most DTT activities reported in the published literature. This means that the DTT activities reported here cannot necessarily be directly compared to generally published results, unless some type of conversion factor is given. This should be noted in the paper, discussed in detail (ie, difference in protocols explained), and optimally, a conversion factor given

based on further experimental work by the authors. Finally, despite a large contribution, there really is no explanation or in depth discussion of why the secondary PMF factor contributes most to DTT activity in summer.

Specific Comments:

Discuss possibility that DTPA not only reduces contributions of metals to DTT activity, but also additional species, such as quinones. Could this result in significant under-measurement of DTT activity in this work? What was the justification for wishing to remove metals contribution to DTT? Are they not part of HULIS, ie possibly as a part of a metal-OM complex?

Lines 116 to 120. This paragraph is not clear. Why should DTT activity be proportional to HULISws concentration with this method. This was a finding reported in the results, but it appears from this paragraph that it simply results from the method used. Clarification is needed.

Somewhere explicitly define the difference between total HULIS, HULIS and HULIS-associated DTT. I assume HULIS-associated DTT is just the DTT activity of the HULIS sample? Sometimes, just the term HULIS is used, which adds to the confusion. Is this total HULIS?. Consistency throughout the paper is needed. Suggest call it HULISws mass concentration and DTT activity of HULISws to distinguish the two.

The PMF factors need clarification. Vehicle emissions apparently include POA (primary) and SOA from vehicle emissions, maybe this should be noted in the figs? Is secondary aerosol mainly biogenic SOA, or are there anthropogenic species contributing to it? If both, can they be separated? Why is there little sulfate in coal combustion? Are there secondary species in biomass burning?

The discussion of possible reasons why DTT activities of the secondary factor are so high in summer needs more attention. Are the authors saying that biogenic SOA is the cause? There are papers that make this dubious claim (Kramer, A., W. Rattanavaraha,

Z. Zhang, A. Gold, J. D. Surratt, and Y.-H. Lin (2016), Assessing the oxidative potential of isoprene-derived epoxides and secondary organic aerosol, Atmos. Env., 130, 211-218.). Given that most sulfate is in this factor (and surprisingly in contrast to the coal combustion factor), it seems that the factor is really all about oxidation processes (as the name for the factor implies). One explanation is that this factor really has contributions from all source, such as coal, vehicles, and even biomass burning, given the loss of levoglucosan through oxidation (although this source is lower in summer). Verma et al, 2015a noted the strong dependence of aging on enhanced quinone DTT activity. This factor may just represent this process. Despite significant discussion of the other factors, this factor is not considered sufficiently give the large role it plays in summer DTT activities, a major finding of the paper.

Lines 271 and on. DTT per HULIS mass is reported. This is interesting, but also interesting would be DTT activities per OM. One could also expand the comparisons of these types of numbers from this study to the large list reported in a recent paper ( Shiraiwa et al., 2017), Aerosol health effects from molecular to global scales, Envir. Sci Technol. , 51, 13545-13567). Make sure to note differences in DTT analytical methods when doing the comparison (eg, no metals in this work).

Regarding trash burning and DTT activities, also see: Vreeland, H., J. J. Schauer, A. G. Russell, J. D. Marshall, A. Fushimi, G. Jain, K. Sethuraman, S. N. Tripathi, and M. H. Bergin (2016), Chemical characterization and toxicity of particulate matter emissions from roadside trash combustion in urban India, Atmos. Env., 147, 22-30.
* * *

---

## Author Comment (AC2) · 6 Mar 2018

1. This paper is of general interest. It provides insights on sources of DTT activity associated with organic species in Beijing.

Author Response: Thanks for the valuable comments.

2. There are, however, some major issues to address. First, the grammar is an issue; the paper needs editing.

Author Response: The paper has been carefully revised and the grammar issue has been addressed.

[Figure]

3. Secondly, unfortunately the authors apparently quantified DTT activities using a non-standard method, ie that of Li et al., 2009, which differs from the original DTT protocol described by Cho et al. (2005). The Cho method is widely utilized and is the basis for most DTT activities reported in the published literature. This means that the DTT activities reported here cannot necessarily be directly compared to generally published results, unless some type of conversion factor is given. This should be noted in the paper, discussed in detail (ie, difference in protocols explained), and optimally, a conversion factor given based on further experimental work by the authors.

Author Response: The major difference between the DTT protocol used by Cho et al. (2005) and Li et al. (2009) is that in Cho's method, at the designed incubation time, trichloroacetic acid was added to quench the reaction. However, Li et al. (2009) found the reaction between DTNB and DTT was very fast, so the absorption reached its maximum value immediately and were very stable for more than 2 hours. Thus, Li et al. (2009) modified Cho's method by eliminating the quenching step. Actually, many recent papers have adopted the modified DTT procedure developed by Li et al. (2009), such as Rattanavaraha et al. (2011), Janssen et al. (2014), Kramer et al. (2016), and Xiong et al. (2017). A few studies have examined the DTT activity of HULISWS (Dou et al., 2015; Lin and Yu, 2011), and they also used a DTT protocol based on Li et al.'s method (2009). Therefore, to let our results be directly comparable to those reported in previous studies, we followed the same protocol of DTT assay as described in Dou et al. (2015) and Lin and Yu (2011).

4. Finally, despite a large contribution, there really is no explanation or in depth discussion of why the secondary PMF factor contributes most to DTT activity in summer.

Author Response: This is mainly due to leading contribution (50.1%) of secondary aerosol formation to HULISWS in summer. As we discussed in detail in the revised manuscript, line 258-270: "In addition to the four combustion-related sources, one secondary source was apportioned by PMF, contributing 30.1% of HULISWS throughout the year. MLR analysis was conducted to evaluate the effects of O3, NOX, SO42−,

Hp+, and LWCp on the secondary formation of HULISWS (HULISWS_SEC). Sulfate was found to be the most significant factor with a regression coefficient of 0.066 (Table S4 in the Supplementary Material). This may be due to the predominant role of sulfate in the particle-phase formation of organosulfates, one important HULISWS component (Xu et al., 2015), through both nucleophilic addition reactions and the salting-in effect (Lin et al., 2012; Riva et al., 2015). Results from the MLR analysis also indicated that an increase of 1 $\mu$g m$-$3 O3 led to an increase of 0.028 $\mu$g m$-$3 HULISWS_SEC. Gaseous highly oxidized multifunctional organic compounds (HOMs) were characterized in the ozonolysis of $\alpha$-pinene in smog chamber experiments (Zhang et al., 2015). It was suggested that, after partitioning to the particle phase, these HOMs could undergo rapid accretion reactions to form oligomers containing multiple carboxylic acid and ester groups, which served as good HULISWS candidates. Considering the higher concentrations of O3 in the non-heating season (Figure S7 in the Supplementary Material), together with higher biogenic VOCs emissions and temperature as well as more intense solar radiation, a larger amount of HULISWS_SEC was produced in the non-heating season (2.01 $\mu$g m$-$3) than in the heating season (1.41 $\mu$g m$-$3)".

We have also had more discussion on the contribution of secondary formation to DTT activity of HULISWS in the revised manuscript, line 295-312: "Secondary formation was the most important source for the extrinsic DTT activity of HULISWS in the non-heating season (44.1%, 0.015 nmol min$-$1 m$-$3) and the second largest contributor throughout the year (25.3%, 0.013 nmol min$-$1 m$-$3). A few smog chamber experiments have been carried out to investigate the ROS activity of SOA from various hydrocarbon precursors, and the intrinsic DTT activity values of several biogenic SOA systems(i.e. isoprene, $\alpha$-pinene, and $\beta$-caryophyllene) were found to be within the range of 2 to 30 pmol min$-$1 per $\mu$g SOA (Kramer et al., 2016; Tuet et al., 2017). Tuet et al. (2017) also observed a much higher intrinsic DTT activity of naphthalene SOA than that of biogenic SOA, and suggested that this was probably due to the aromatic species, especially nitroaromatics, in naphthalene SOA. The intrinsic DTT activity of HULISWS_SEC measured in this study is 7.45 pmol min$-$1 per $\mu$g HULISWS_SEC,

which is within the reported intrinsic DTT activity range of biogenic SOA. Moreover, results from MLR analysis indicated that both sulfate and ozone were positively correlated with HULISWS_SEC (Table S2), suggesting that HULISWS resolved in this factor could mainly consist of some less ROS-active SOA components, such as organosulfates (Chen et al., 2011; Lin et al., 2012). Although chamber experiments reported the formation of ROS-active HOMs or organic peroxides through the ozonolysis of biogenic VOCs (Docherty et al., 2005; Zhang et al., 2015), the production yields of these peroxides were generally low and thus could not have a major influence on the DTT activity of HULISWS_SEC. However, since secondary formation predominated in HULISWS formation (Figure 4A), especially in the non-heating season (50.1%), even with a lower intrinsic DTT activity, secondary aerosol formation still serves as a significant contributor to HULISWS-associated redox activity in Beijing. It should be noted that the contributions of secondary formation processes to both HULISWS and DTT activity of HULISWS could even have been underestimated in this study, because HULISWS secondarily formed through the aging of biomass burning and vehicle emissions was resolved in factor B and D and could not be accurately quantified".

Specific Comments:

5. Discuss possibility that DTPA not only reduces contributions of metals to DTT activity, but also additional species, such as quinones. Could this result in significant under-measurement of DTT activity in this work? What was the justification for wishing to remove metals contribution to DTT? Are they not part of HULIS, ie possibly as a part of a metal-OM complex?

Author Response: In this paper, we meant to measure the DTT activity of organic HULISWS. However, even after HLB separation, there are still a small amount of metal retained in the HULISWS fraction (e.g. Cu: 17%, Fe: 10%, Zn: <5%; Lin and Yu, 2011). Based on previous studies, some metals (e.g. Cu) could cause considerable DTT consumption yet contribute negligibly to ROS generation, some (e.g. Fe) are inactive in DTT oxidation (Xiong et al., 2017), and some could form complex with DTT and

cause false positive DTT activity (Kreel et al., 2001). So in the assessment of oxidative potential of organic compounds, many papers added EDTA as the metal chelator in the incubation solution at the beginning of DTT experiment to eliminate the influence of metals (Kramer et al., 2016; Rattanavaraha et al., 2011). However, Charrier et al. (2015) found EDTA could not only chelate metal but also significantly suppress the DTT response of quinone by $\sim$80%, and this could lead to a large system error in DTT experiment. Besides EDTA, DTPA is another common metal chelator. Although DTPA could also suppress the DTT response of quinone by about 20% (Dou et al., 2015), in order to eliminate the influence of metals and decrease the system error in DTT assay, Lin and Yu, (2011) chose DTPA instead of EDTA as metal chelator. Moreover, considering this method is widely accepted in HULISWS redox activity measurement, we adopt the same method in order to be consistent and make comparison with other HULISWS studies (Dou et al., 2015; Lin and Yu, 2011; Wang et al., 2017). In the revised version, we mentioned the DTT activity of HULISWS measured was underestimated and could not be directly comparable with that does not use DTPA as metal chelators.

In the revised manuscript, line 126-128, we've added: "Since DTPA was added to suppress DTT consumption by metals ions throughout the incubation process and may affect the DTT response of quinones (Dou et al., 2015), the DTT activity of HULISWS measured here may be underestimated and is not directly comparable to those using DTT assay conducted without DTPA".

6. Lines 116 to 120. This paragraph is not clear. Why should DTT activity be proportional to HULISws concentration with this method. This was a finding reported in the results, but it appears from this paragraph that it simply results from the method used. Clarification is needed.

Author Response: Thanks for this comments. Yes, the result that DTT activity is proportional to HULISWS concentration is derived from the method we used. In our paper, the incubation time (90 min) of DTT assay fell into the linear time-dependent range. In another word, the catalytic reaction rate is constant, that is, the reaction is zero-order

with respect to DTT (Lin and Yu, 2011). Thus, the catalytic DTT oxidation rate is proportional to the concentration of catalyst, that is, the abundance of DTT active moieties in HULISWS. In the results and discussion part, we meant to report the good correlation between HULISWS and DTT activity to further clarify that our measurement of DTT activity of HULISWS fell into the linear range. In order to eliminate misleading, we now modified our expression in the revised version.

In the revised manuscript, line 116-119, we also added: "Previous study observed that the time-dependent consumption of DTT catalysed by HULISWS is linear when DTT consumption is less than 90% (Lin and Yu, 2011). We have also examined the HULISWS-catalysed DTT consumption as a function of time and obtained a similar result as Lin and Yu (2011). In our study, the HULISWS-catalysed DTT consumption of all 66 samples are between 3.6% and 77.0%, and the measured DTT activity is linearly proportional to HULISWS mass concentration."

7. Somewhere explicitly define the difference between total HULIS, HULIS and HULIS-associated DTT. I assume HULIS-associated DTT is just the DTT activity of the HULIS sample? Sometimes, just the term HULIS is used, which adds to the confusion. Is this total HULIS? Consistency throughout the paper is needed. Suggest call it HULISws mass concentration and DTT activity of HULISws to distinguish the two.

Author Response: This is a nice suggestion. Total HULISWS and HULIS-associated DTT have been revised to HULISws mass concentration and DTT activity of HULISws in the revised version accordingly.

8. The PMF factors need clarification. Vehicle emissions apparently include POA (primary) and SOA from vehicle emissions, maybe this should be noted in the figs?

Author Response: It is true that both vehicle emissions and biomass burning factors contain HULISWS secondarily formed from the aging process, and we've discussed them in the manuscript. As in the figure, we prefer to keep it unchanged.

9. Is secondary aerosol mainly biogenic SOA, or are there anthropogenic species contributing to it? If both, can they be separated?

Author Response: In Figure 3, we could see that some anthropogenic SOA markers (e.g. 1,2,3-benzenetricarboxylic acid and 1,2,4-benzenetricarboxylic acid) were also resolved in this secondary factor. So this factor is a mixed source of both biogenic and anthropogenic SOA. But we think it is probably dominated by biogenic SOA, and it is difficult to differentiate the biogenic SOA and anthropogenic SOA in this study due to the lack of appropriate anthropogenic SOA markers. Moreover, as we discussed in detail in the manuscript, some anthropogenic SOA in HULISWS, which were formed from the aging of biomass burning and vehicle emissions, were resolved in the biomass burning and vehicle factors instead of this secondary factor, and could not be accurately quantified again due to the lack of appropriate anthropogenic SOA markers.

10. Why is there little sulfate in coal combustion?

Author Response: Since most sulfate ($\sim$70%) was assigned to the secondary PMF factor, there is only little sulfate in coal combustion factor. Results from previous PMF analysis of Beijing PM2.5 samples showed that secondary sulfate factor was always well separated from the coal combustion factor (e.g. Song et al. (2006); Yu et al. (2013); Zhang et al. (2013)), and our result is consistent with those from previous source apportionment studies.

11. Are there secondary species in biomass burning?

Author Response: Yes, small fractions of 4M5NC and 3M6NC were resolved in the biomass burning factor, which indicated that SOA from aging of biomass burning might also be resolved in this factor.

12. The discussion of possible reasons why DTT activities of the secondary factor are so high in summer needs more attention. Are the authors saying that biogenic SOA is the cause? There are papers that make this dubious claim (Kramer, A., W.

Rattanavaraha, Z. Zhang, A. Gold, J. D. Surratt, and Y.-H. Lin (2016), Assessing the oxidative potential of isoprene-derived epoxides and secondary organic aerosol, Atmos. Env., 130, 211- 218.).

Author Response: Please refer to the above response to comment 4.

13. Given that most sulfate is in this factor (and surprisingly in contrast to the coal combustion factor), it seems that the factor is really all about oxidation processes (as the name for the factor implies). One explanation is that this factor really has contributions from all source, such as coal, vehicles, and even biomass burning, given the loss of levoglucosan through oxidation (although this source is lower in summer). Verma et al, 2015a noted the strong dependence of aging on enhanced quinone DTT activity. This factor may just represent this process. Despite significant discussion of the other factors, this factor is not considered sufficiently give the large role it plays in summer DTT activities, a major finding of the paper.

Author Response: Please refer to our response to comment 4.

14. Lines 271 and on. DTT per HULIS mass is reported. This is interesting, but also interesting would be DTT activities per OM. One could also expand the comparisons of these types of numbers from this study to the large list reported in a recent paper ( Shiraiwa et al., 2017), Aerosol health effects from molecular to global scales, Envir. Sci Technol. , 51, 13545-13567). Make sure to note differences in DTT analytical methods when doing the comparison (eg, no metals in this work).

Author Response: In this paper, we only measured the DTT activity to HULISWS fraction instead of the whole organic matter (OM). So it does not make much sense to normalize the DTT activity obtained in this study by the mass of OM. Our major objective is to explore the oxidative potential of HULISWS in the atmosphere and make comparisons to the redox activity value of HULISWS reported by previous studies. Since HULISWS is just a fraction of OM, results obtained in this study are not comparable to those obtained on the OM or total PM2.5 extract.

15. Regarding trash burning and DTT activities, also see: Vreeland, H., J. J. Schauer, A. G. Russell, J. D. Marshall, A. Fushimi, G. Jain, K. Sethuraman, S. N. Tripathi, and M. H. Bergin (2016), Chemical characterization and toxicity of particulate matter emissions from roadside trash combustion in urban India, Atmos. Env., 147, 22-30.

Author Response: Thanks for this suggestion and we have read the paper carefully. The paper above did not mention the detailed DTT protocol they used. Considering they mentioned Cho's method, we assume they did not add metal chelators during the incubation process. However, when using water to extract PM2.5 collected on quartz filters, not only WSOM but also water-soluble metals could be extracted. Therefore, the DTT activity reported by them was induced by both WSOM and metals. This makes it incomparable to our result.

Reference:

Charrier, J. G., Richards-Henderson, N. K., Bein, K. J., McFall, A. S., Wexler, A. S. and Anastasio, C.: Oxidant production from source-oriented particulate matter - Part 1: Oxidative potential using the dithiothreitol (DTT) assay, Atmos. Chem. Phys., 15(5), 2327–2340, 2015. Cho, A. K., Sioutas, C., Miguel, A. H., Kumagai, Y., Schmitz, D. A., Singh, M., Eiguren-Fernandez, A. and Froines, J. R.: Redox activity of airborne particulate matter at different sites in the Los Angeles Basin, Environ. Res., 99(1), 40–47, 2005. Dou, J., Lin, P., Kuang, B. and Yu, J. Z.: Reactive oxygen species production mediated by humic-like substances in atmospheric aerosols: Enhancement effects by pyridine, imidazole, and their derivatives, Environ. Sci. Technol., 49(11), 6457–6465, 2015. Janssen, N. A. H., Yang, A., Strak, M., Steenhof, M., Hellack, B., Gerlofs-Nijland, M. E., Kuhlbusch, T., Kelly, F., Harrison, R., Brunekreef, B., Hoek, G. and Cassee, F.: Oxidative potential of particulate matter collected at sites with different source characteristics, Sci. Total Environ., 472, 572–581, 2014. Kramer, A. J., Rattanavaraha, W., Zhang, Z., Gold, A., Surratt, J. D. and Lin, Y. H.: Assessing the oxidative potential of isoprene-derived epoxides and secondary organic aerosol, Atmos. Environ., 130, 211–218, 2016. Kreel, A., Leniak, W., Jeowska-Bojczuk, M., Mlynarz, P., Brasu, J.,

Kozlowski, H. and Bal, W.: Coordination of heavy metals by dithiothreitol, a commonly used thiol group protectant, J. Inorg. Biochem., 84(1–2), 77–88, 2001. Li, Q., Wyatt, A. and Kamens, R. M.: Oxidant generation and toxicity enhancement of aged-diesel exhaust, Atmos. Environ., 43(5), 1037–1042, 2009. Lin, P. and Yu, J. Z.: Generation of reactive oxygen species mediated by humic-like substances in atmospheric aerosols, Environ. Sci. Technol., 45(24), 10362–10368, 2011. Rattanavaraha, W., Rosen, E., Zhang, H., Li, Q., Pantong, K. and Kamens, R. M.: The reactive oxidant potential of different types of aged atmospheric particles: An outdoor chamber study, Atmos. Environ., 45(23), 3848–3855, 2011. Song, Y., Zhang, Y., Xie, S., Zeng, L., Zheng, M., Salmon, L. G., Shao, M. and Slanina, S.: Source apportionment of PM2.5 in Beijing by positive matrix factorization, Atmos. Environ., 40(8), 1526–1537, 2006. Tuet, W. Y., Chen, Y., Xu, L., Fok, S., Gao, D., Weber, R. J. and Ng, N. L.: Chemical oxidative potential of secondary organic aerosol (SOA) generated from the photooxidation of biogenic and anthropogenic volatile organic compounds, Atmos. Chem. Phys., 17(2), 839–853, 2017. Wang, Y., Hu, M., Lin, P., Guo, Q., Wu, Z., Li, M., Zeng, L., Song, Y., Zeng, L., Wu, Y., Guo, S., Huang, X. and He, L.: Molecular characterization of nitrogen-containing organic compounds in humic-like substances emitted from straw residue burning, Environ. Sci. Technol., 51(11), 5951–5961, 2017. Xiong, Q., Yu, H., Wang, R., Wei, J. and Verma, V.: Rethinking dithiothreitol-based particulate matter oxidative potential : Measuring dithiothreitol consumption versus reactive oxygen species generation, Environ. Sci. Technol., 51(11), 6507–6514, 2017. Yu, L., Wang, G., Zhang, R., Zhang, L., Song, Y., Wu, B., Li, X., An, K. and Chu, J.: Characterization and source apportionment of PM2.5 in an urban environment in Beijing, Aerosol Air Qual. Res., 13(2), 574–583, 2013. Zhang, R., Jing, J., Tao, J., Hsu, S. C., Wang, G., Cao, J., Lee, C. S. L., Zhu, L., Chen, Z., Zhao, Y. and Shen, Z.: Chemical characterization and source apportionment of PM2.5 in Beijing: Seasonal perspective, Atmos. Chem. Phys., 13(14), 7053–7074, 2013.

[Figure]

2017.

---

## Author Response (AR1)

**Point-by-point Response to Review Comments on "Sources and oxidative potential of water-soluble humic-like substances (HULISWS) in fine particulate matter (PM2.5) in Beijing"**

**Reviewer #2:**

It's well known that the toxicity of PM2.5 is greatly related to its chemical composition and pollution sources. This work analyzed PM2.5 samples collected in Beijing during a one-year period, and the levels and oxidative potential (indicated by DTT) of a major water-soluble PM2.5 component, i.e. water-soluble humic-like substances (HULISWS), were reported. With the aid of various characteristic source tracers, PMF was applied to apportion the major sources of both HULISWS and its associated DTT activity. There are some new and interesting findings. The major sources of both HULISWS and DTT activity were coal combustion, biomass burning, traffic exhaust, waste incineration, and secondary formation. Waste incineration was probably identified as a contributor to HULISWS for the first time. Moreover, HULISWS from vehicle emissions was found as the most ROS-active, and HULISWS from secondary aerosol formation showed a lower intrinsic DTT ability than those of most primary sources except for coal combustion. This paper is well-written. The study is clear, informative, and novel in general, and the major data and their interpretation are scientifically sound. I suggest it to be considered by ACP for publication if the following concerns could be addressed.

**Author Response: Thanks for the comments.**

1. Line 22: Is it necessary to define the waste as "plastic waste", as terephthalic acid is a marker of plastics? **Author Response:** Since terephthalic acid is an important industrial material for making PET (polyethylene terephthalate) plastics, they have been found to be abundant in plastic burning smokes. In developing countries, plastic materials are dumped as domestic waste, which makes the waste plastic-enriched. In this case, terephthalic acid serves as a marker of plastic burning and plastic-enriched domestic waste burning. Therefore, we will keep the name of this factor as "waste incineration".

2. Line 39-40: Add "an" before electron and "a" before continuous.

Author Response: This revision has been done.

In the revised manuscript, line 40-41 (in the manuscript file of "BJ HULIS\_clear"; the same below): "The reversible redox sites in HULISWS fraction could serve as an electron transfer intermediate and lead to continuous production of ROS (Lin and Yu, 2011)."

3. Line 71: If the samples were taken every 6 days for a one-year period, there should be 60 samples. Why there were 66 samples as listed in line 123?

Author Response: In winter time, we collected several additional samples during severe polluted periods.

In the revised manuscript, line 70-71, we've added "...with several additional samples collected during severe polluted periods".

4. Line 100: How many individual hopanes have been identified? I guess the input species "hopane" in PMF must be the sum of all identified hopanes, right? What are the concentrations of hopanes? What are the water-soluble ions identified? The author may need to include a table in the supplementary information that provide levels of hopanes, water-soluble ions, EC and OC in the batch of  $PM_{2.5}$  samples analyzed.

**Author Response:** Five individual hopanes were identified, and these 5 hopanes were lumped together and put into PMF analysis. The concentrations of the measured hopanes, ions, EC and OC were listed in Table S1 in the revised supplementary material.

In the revised manuscript, line 103-104, we've added: "Concentrations of hopanes, levoglucosan, watersoluble ions, EC and OC were listed in Table S1 in the Supplementary Material."

5. Line 108: DTPA was spiked to chelate transition metals. Could it also affect or even remove some HULISws components?

Author Response: In the previous work by Lin and Yu (2011), they assessed the DTT consumption by 1) HULISWS sample without DTPA; 2) HULISWS samples with DTPA; and 3) standard mixture solution of metals with concentrations similar to those measured in HULISWS fraction. They found that the DTT consumption by residue metals in the absence of DTPA is similar to the difference between DTT consumption by equivalent amount of HULISWS in the absence and presence of DTPA. Their result suggested that the DTPA spiked to chelate transition metals could barely affect the HULISWS components. Although Dou et al. (2015) found that DTPA could suppress the DTT response of quinone by about 20%, considering this method is widely accepted in HULISWS redox activity measurement, we adopt the same

method in order to be consistent and make comparison with other  $HULIS_{WS}$  studies (Dou et al., 2015; Lin and Yu, 2011). In the revised version, we mentioned the DTT activity of  $HULIS_{WS}$  measured was underestimated and could not be directly comparable with that does not use DTPA as metal chelators.

In the revised manuscript, line 126-128, we've added: "Since DTPA was added to suppress DTT consumption by metals ions throughout the incubation process and may affect the DTT response of quinones (Dou et al., 2015), the DTT activity of HULISWS measured here may be underestimated and is not directly comparable to those studies conducting DTT assay experiments without DTPA."

6. Lines 130 to 131: Were all the reference data observed during a one-year period and comparable to present study?

Author Response: Data reported for Guangzhou and Nansha sites were the average concentration of samples collected during a one-year period (2009 January – 2009 December). For data measured in Lanzhou, they were the average concentration of samples collected during two representative periods (June-July in summer and December in winter during 2012-2013).

**Reference:**

- Dou, J., Lin, P., Kuang, B., Yu, J.Z., 2015. Reactive oxygen species production mediated by humic-like substances in atmospheric aerosols: Enhancement effects by pyridine, imidazole, and their derivatives. Environ. Sci. Technol. 49, 6457–6465.
- Lin, P., Yu, J.Z., 2011. Generation of reactive oxygen species mediated by Humic-like substances in atmospheric aerosols. Environ. Sci. Technol. 45, 10362–10368.

**Reviewer #3:**

1. This paper is of general interest. It provides insights on sources of DTT activity associated with organic species in Beijing.

Author Response: Thanks for the valuable comments.

2. There are, however, some major issues to address. First, the grammar is an issue; the paper needs editing.

Author Response: The paper has been carefully revised and the grammar issue has been addressed.

3. Secondly, unfortunately the authors apparently quantified DTT activities using a non-standard method, ie that of Li et al., 2009, which differs from the original DTT protocol described by Cho et al. (2005). The Cho method is widely utilized and is the basis for most DTT activities reported in the published literature. This means that the DTT activities reported here cannot necessarily be directly compared to generally published results, unless some type of conversion factor is given. This should be noted in the paper, discussed in detail (ie, difference in protocols explained), and optimally, a conversion factor given based on further experimental work by the authors.

**Author Response:** The major difference between the DTT protocol used by Cho et al. (2005) and Li et al. (2009) is that in Cho's method, at the designed incubation time, trichloroacetic acid was added to quench the reaction. However, Li et al. (2009) found the reaction between DTNB and DTT was very fast, so the absorption reached its maximum value immediately and were very stable for more than 2 hours. Thus, Li et al. (2009) modified Cho's method by eliminating the quenching step. Actually, many recent papers have adopted the modified DTT procedure developed by Li et al. (2009), such as Rattanavaraha et al. (2011), Janssen et al. (2014), Kramer et al. (2016), and Xiong et al. (2017). A few studies have examined the DTT activity of HULISWS (Dou et al., 2015; Lin and Yu, 2011), and they also used a DTT protocol based on Li et al.'s method (2009). Therefore, to let our results be directly comparable to those reported in previous studies, we followed the same protocol of DTT assay as described in Dou et al. (2015) and Lin and Yu (2011).

4. Finally, despite a large contribution, there really is no explanation or in depth discussion of why the secondary PMF factor contributes most to DTT activity in summer.

Author Response: This is mainly due to leading contribution (50.1%) of secondary aerosol formation to HULISWS in summer. As we discussed in detail in the revised manuscript, line 257-269 (in the manuscript file of "BJ HULIS\_clear"; the same below): "In addition to the four combustion-related sources, one secondary source was apportioned by PMF, contributing 30.1% of HULISWS throughout the year. MLR analysis was conducted to evaluate the effects of  $O_3$ ,  $NO_X$ ,  $SO_4^{2-}$ ,  $H_p^+$ , and  $LWC_p$  on the secondary formation of HULISWS (HULISWS-SEC). Sulfate was found to be the most significant factor with a regression coefficient of 0.066 (Table S4 in the Supplementary Material). This may be due to the predominant role of sulfate in the particle-phase formation of organosulfates, one important HULISWS component (Xu et al., 2015), through both nucleophilic addition reactions and the salting-in effect (Lin et al., 2012; Riva et al., 2015). Results from the MLR analysis also indicated that an increase of 1  $\mu$ g m-3 O3 led to an increase of 0.028  $\mu$ g m-3 HULISWS SEC. Gaseous highly oxidized multifunctional organic compounds (HOMs) were characterized in the ozonolysis of  $\alpha$ -pinene in smog chamber experiments (Zhang et al., 2015). It was suggested that, after partitioning to the particle phase, these HOMs could undergo rapid accretion reactions to form oligomers containing multiple carboxylic acid and ester groups, which served as good HULISWS candidates. Considering the higher concentrations of O3 in the non-heating season (Figure S7 in the Supplementary Material), together with higher biogenic VOCs emissions and temperature as well as more intense solar radiation, a larger amount of HULISWS SEC was produced in the non-heating season (2.01  $\mu$ g m-3) than in the heating season (1.41  $\mu$ g m-3). "

We have also had more discussion on the contribution of secondary formation to DTT activity of HULISWS in the revised manuscript, line 294-311: "Secondary formation was the most important source for the extrinsic DTT activity of HULISWS in the non-heating season (44.1%, 0.015 nmol min-1 m-3) and the second largest contributor throughout the year (25.3%, 0.013 nmol min-1 m-3). A few smog chamber experiments have been carried out to investigate the ROS activity of SOA from various hydrocarbon precursors, and the intrinsic DTT activity values of several biogenic SOA systems (i.e. isoprene,  $\beta$ -pinene, and  $\alpha$ -caryophyllene) were found to be within the range of 2 to 30 pmol min-1 per µg SOA (Kramer et al., 2016; Tuet et al., 2017). Tuet et al. (2017) also observed a much higher intrinsic DTT activity of naphthalene SOA than that of biogenic SOA, and suggested that this was probably due to the aromatic species, especially nitroaromatics, in naphthalene SOA. The intrinsic DTT activity of HULISWS\_SEC measured in this study is 7.45 pmol min-1 per µg HULISWS\_SEC, which is within the reported intrinsic DTT activity or correlated with HULISWS\_SEC (Table S4), suggesting that HULISWS resolved in this factor could mainly consist of some less ROS-active SOA components, such as organosulfates (Chen et al., 2011; Lin et al., 2012). Although chamber experiments reported the formation of ROS-active HOMs or organic

peroxides through the ozonolysis of biogenic VOCs (Docherty et al., 2005; Zhang et al., 2015), the production yields of these peroxides were generally low and thus could not have a major influence on the DTT activity of HULISWS\_SEC. However, since secondary formation predominated in HULISWS formation (Figure 4A), especially in the non-heating season (50.1%), even with a lower intrinsic DTT activity, secondary aerosol formation still serves as a significant contributor to HULISWS-associated redox activity in Beijing. It should be noted that the contributions of secondary formation processes to both HULISWS and DTT activity of HULISWS could even have been underestimated in this study, because HULISWS secondarily formed through the aging of biomass burning and vehicle emissions was resolved in factor B and D and could not be accurately quantified."

**Specific Comments:**

5. Discuss possibility that DTPA not only reduces contributions of metals to DTT activity, but also additional species, such as quinones. Could this result in significant under-measurement of DTT activity in this work? What was the justification for wishing to remove metals contribution to DTT? Are they not part of HULIS, ie possibly as a part of a metal-OM complex?

Author Response: In this paper, we meant to measure the DTT activity of organic HULISWS. However, even after HLB separation, there are still a small amount of metal retained in the HULISWS fraction (e.g. Cu: 17%, Fe: 10%, Zn: <5%; Lin and Yu, 2011). Based on previous studies, some metals (e.g. Cu) could cause considerable DTT consumption yet contribute negligibly to ROS generation, some (e.g. Fe) are inactive in DTT oxidation (Xiong et al., 2017), and some could form complex with DTT and cause false positive DTT activity (Kreel et al., 2001). So in the assessment of oxidative potential of organic compounds, many papers added EDTA as the metal chelator in the incubation solution at the beginning of DTT experiment to eliminate the influence of metals (Kramer et al., 2016; Rattanavaraha et al., 2011). However, Charrier et al. (2015) found EDTA could not only chelate metal but also significantly suppress the DTT response of quinone by ~80%, and this could lead to a large system error in DTT experiment. Besides EDTA, DTPA is another common metal chelator. Although DTPA could also suppress the DTT response of quinone by about 20% (Dou et al., 2015), in order to eliminate the influence of metals and decrease the system error in DTT assay, Lin and Yu, (2011) chose DTPA instead of EDTA as metal chelator. Moreover, considering this method is widely accepted in HULISWS redox activity measurement, we adopt the same method in order to be consistent and make comparison with other HULISWS studies (Dou et al., 2015; Lin and Yu, 2011; Wang et al., 2017). In the revised version, we mentioned the DTT

activity of  $HULIS_{WS}$  measured was underestimated and could not be directly comparable with that does not use DTPA as metal chelators.

In the revised manuscript, line 126-128, we've added: "Since DTPA was added to suppress DTT consumption by metals ions throughout the incubation process and may affect the DTT response of quinones (Dou et al., 2015), the DTT activity of HULISWS measured here may be underestimated and is not directly comparable to those studies conducting DTT assay experiments without DTPA."

6. Lines 116 to 120. This paragraph is not clear. Why should DTT activity be proportional to HULISws concentration with this method. This was a finding reported in the results, but it appears from this paragraph that it simply results from the method used. Clarification is needed.

Author Response: Thanks for this comments. Yes, the result that DTT activity is proportional to  $HULIS_{WS}$  concentration is derived from the method we used. In our paper, the incubation time (90 min) of DTT assay fell into the linear time-dependent range. In another word, the catalytic reaction rate is constant, that is, the reaction is zero-order with respect to DTT (Lin and Yu, 2011). Thus, the catalytic DTT oxidation rate is proportional to the concentration of catalyst, that is, the abundance of DTT active moieties in  $HULIS_{WS}$ . In the results and discussion part, we meant to report the good correlation between  $HULIS_{WS}$  and DTT activity to further clarify that our measurement of DTT activity of  $HULIS_{WS}$  fell into the linear range. In order to eliminate misleading, we now modified our expression in the revised version.

In the revised manuscript, line 116-119, we also added: "Previous study observed that the time-dependent consumption of DTT catalysed by  $HULIS_{WS}$  was linear when DTT consumption was less than 90% (Lin and Yu, 2011). We have also examined the  $HULIS_{WS}$ -catalysed DTT consumption as a function of time and obtained a similar result as Lin and Yu (2011). In our study, the  $HULIS_{WS}$ -catalysed DTT consumption of all 66 samples were between 3.6% and 77.0%, and the measured DTT activity was linearly proportional to  $HULIS_{WS}$  mass concentration."

7. Somewhere explicitly define the difference between total HULIS, HULIS and HULIS- associated DTT. I assume HULIS-associated DTT is just the DTT activity of the HULIS sample? Sometimes, just the term HULIS is used, which adds to the confusion. Is this total HULIS? Consistency throughout the paper is needed. Suggest call it HULIS mass concentration and DTT activity of HULIS we to distinguish the two.

Author Response: This is a nice suggestion. Total  $HULIS_{WS}$  and HULIS-associated DTT have been revised to HULIS mass concentration and DTT activity of HULIS in the revised version accordingly.

8. The PMF factors need clarification. Vehicle emissions apparently include POA (primary) and SOA from vehicle emissions, maybe this should be noted in the figs?

Author Response: It is true that both vehicle emissions and biomass burning factors contain  $HULIS_{WS}$  secondarily formed from the aging process, and we've discussed them in the manuscript. As in the figure, we prefer to keep it unchanged.

9. Is secondary aerosol mainly biogenic SOA, or are there anthropogenic species contributing to it? If both, can they be separated?

**Author Response:** In Figure 3, we could see that some anthropogenic SOA markers (e.g. 1,2,3benzenetricarboxylic acid and 1,2,4-benzenetricarboxylic acid) were also resolved in this secondary factor. So this factor is a mixed source of both biogenic and anthropogenic SOA. But we think it is probably dominated by biogenic SOA, and it is difficult to differentiate the biogenic SOA and anthropogenic SOA in this study due to the lack of appropriate anthropogenic SOA markers. Moreover, as we discussed in detail in the manuscript, some anthropogenic SOA in HULISWS, which were formed from the aging of biomass burning and vehicle emissions, were resolved in the biomass burning and vehicle factors instead of this secondary factor, and could not be accurately quantified again due to the lack of appropriate anthropogenic SOA markers.

**10. Why is there little sulfate in coal combustion?**

Author Response: Since most sulfate (~70%) was assigned to the secondary PMF factor, there is only little sulfate in coal combustion factor. Results from previous PMF analysis of Beijing  $PM_{2.5}$  samples showed that secondary sulfate factor was always well separated from the coal combustion factor (e.g. Song et al. (2006); Yu et al. (2013); Zhang et al. (2013)), and our result is consistent with those from previous source apportionment studies.

**11. Are there secondary species in biomass burning?**

**Author Response:** Yes, small fractions of 4M5NC and 3M6NC were resolved in the biomass burning factor, which indicated that SOA from aging of biomass burning might also be resolved in this factor.

12. The discussion of possible reasons why DTT activities of the secondary factor are so high in summer needs more attention. Are the authors saying that biogenic SOA is the cause? There are papers that make this dubious claim (Kramer, A., W. Rattanavaraha, Z. Zhang, A. Gold, J. D. Surratt, and Y.-H. Lin (2016), Assessing the oxidative potential of isoprene-derived epoxides and secondary organic aerosol, Atmos. Env., 130, 211- 218.).

Author Response: Please refer to the above response to comment 4.

13. Given that most sulfate is in this factor (and surprisingly in contrast to the coal combustion factor), it seems that the factor is really all about oxidation processes (as the name for the factor implies). One explanation is that this factor really has contributions from all source, such as coal, vehicles, and even biomass burning, given the loss of levoglucosan through oxidation (although this source is lower in summer). Verma et al, 2015a noted the strong dependence of aging on enhanced quinone DTT activity. This factor may just represent this process. Despite significant discussion of the other factors, this factor is not considered sufficiently give the large role it plays in summer DTT activities, a major finding of the paper.

Author Response: Please refer to our response to comment 4.

14. Lines 271 and on. DTT per HULIS mass is reported. This is interesting, but also interesting would be DTT activities per OM. One could also expand the comparisons of these types of numbers from this study to the large list reported in a recent paper (Shiraiwa et al., 2017), Aerosol health effects from molecular to global scales, Envir. Sci Technol. , 51, 13545-13567). Make sure to note differences in DTT analytical methods when doing the comparison (eg, no metals in this work).

Author Response: In this paper, we only measured the DTT activity to  $HULIS_{WS}$  fraction instead of the whole organic matter (OM). So it does not make much sense to normalize the DTT activity obtained in this study by the mass of OM. Our major objective is to explore the oxidative potential of  $HULIS_{WS}$  in the atmosphere and make comparisons to the redox activity value of  $HULIS_{WS}$  reported by previous studies. Since  $HULIS_{WS}$  is just a fraction of OM, results obtained in this study are not comparable to those obtained on the OM or total  $PM_{2.5}$  extract.

15. Regarding trash burning and DTT activities, also see: Vreeland, H., J. J. Schauer, A. G. Russell, J. D. Marshall, A. Fushimi, G. Jain, K. Sethuraman, S. N. Tripathi, and M. H. Bergin (2016), Chemical

characterization and toxicity of particulate matter emissions from roadside trash combustion in urban India, Atmos. Env., 147, 22-30.

Author Response: Thanks for this suggestion and we have read the paper carefully. The paper above did not mention the detailed DTT protocol they used. Considering they mentioned Cho's method, we assume they did not add metal chelators during the incubation process. However, when using water to extract  $PM_{2.5}$  collected on quartz filters, not only WSOM but also water-soluble metals could be extracted. Therefore, the DTT activity reported by them was induced by both WSOM and metals. This makes it incomparable to our result.

Humic-like substances (HULISws) are a mixture of compounds-that containing polycyclic ring structures with aliphatic side chains and multiple polar functional groups. They account for, and a significant proportion (30%–80%) of the water-soluble organic matter (WSOM) in PM2.5 (Graber and Rudich, 2006; Kuang et al., 2015; Lin et al., 2010a). HULISws have also recently been recognized to be highly redox-active and they play a significant role in driving PM-associated ROS formation (Dou et al., 2015; Lin and Yu, 2011; Verma et al., 2015a). The reversible redox sites in HULISws fraction could serve as an electron transfer intermediatery and lead to continuous production of ROS (Lin and Yu, 2011). Actually, many recent studies have reported the significant role of HULISws in driving PM-associated ROS formation (Dou et al., 2015; Lin and Yu, 2011; Verma et al., 2015a). Dithiothreitol (DTT) assay have been widely used frequently applied to evaluate the oxidative potential capacity of HULISws and PM2.5 components., especially for organic compounds (Xiong et al., 2017). By adopting this method, Verma et al. (2015b) found that HULISws contributedaused approximately 45% of DTT activity of the water extracts from PM2.5 samples collected d in Atlanta, USA4. Th which is was 5% higher than that induced by water-soluble metals (Verma et al., 2015b). Lin and Yu (2011) also Furthermore, found that be DTT activity of HULISws accounted for is about-79%±12% of DTT activity caused by the whole

WSOM fraction in PM2.5 sampleds in Pearl River Delta (PRD) region, China-(Lin and Yu, 2011). G, suggesting a substantial

health threat induced by HULISws. Thus, given the considerable amount of HULISws in PM2.5 and their high-ROS generation ability, both field measurements and smog chamber experiments have been conducted to determine their formation mechanisms pathways and atmospheric-origins in the atmosphere (Kautzman et al., 2010; Lin et al., 2010b; Sato et al., 2012). Biomass burning and secondary formation have been suggested to be the major sources of atmospheric HULISws (Kautzman et al., 2010; Lin et al., 2010b). However, until now, studies on the quantitative source apportionment of HULISws remain relatively rareare still limited (Kuang et al., 2015), and information on the source-specific contribution to their redox activity is lacking.

Beijing, the capital of China located in the North China Plain, is a political and cultural center with an extremely densely population. On the other hand, it has become one of the most polluted cities in the world, with an annual  $PM_{2.5}$  concentration of up to 89.5 µg m-3 in 2013 (Li et al., 2017). Therefore, it presents an ideal location to study the chemical characteristics of HULISWS as well as their sources and potential redox activity.

In this study, our major objective is to investigate the ROS-forming ability of HULISws in relation to different sources and meteorological conditions. Thus, a total of 66 PM2.5 samples collected in Beijing during a 1-year period were analyzed. Concentrations of total HULISws were quantified, together with some characteristic individual HULISws species and the major aerosol components. HULISws-associatedThe redox activity of HULISws 
[revised manuscript text omitted]

The HULISWS-catalysed DTT consumption of each sample was normalized by the volume of air sampled (DTTv, defined as extrinsic DTT activity and expressed in units of nmol min-1 m-3) and the HULISWS mass (DTTm, defined as intrinsic DTT activity and expressed in units of mol min-1 per  $\mu$ g HULISWS) (Dou et al., 2015; Verma et al., 2014), respectively. The mathematical expressions of DTTv and DTTm are shown below.

Extrinsic DTT activity
$$(DTT_V) = \frac{R_{DTT}(\%) \times n_{DTT}(nmol)}{t(min) \times Air \ volumn(m^{-3})}$$
 E.q. (1)

Intrinsic DTT activity
$$(DTT_m) = \frac{DTT_V (nmol \min m^{-3})}{HULIS_{WS} (\mu g m^{-3})}$$
 E.q. (2)

Since DTPA was added to suppress DTT consumption by metals ions throughout the incubation process and may affect the DTT response of quinones (Dou et al., 2015), the DTT activity of  $HULIS_{WS}$  measured here may be underestimated and is not directly comparable to those studies conducting DTT assay experiments without DTPA.

**2.4 Source apportionment**

In this study, the United States Environmental Protection Agency PMF 5.0 was applied to identify and apportion the sources of both-HULISWS and apportion their contributions to both HULISWS and HULISWS associated the extrinsic redox-DTT activity of HULISWS. As suggested by Henry et al. (1984), the minimum sample size of N for PMF analysis was 30 + (V + 3)/2, where V is the number of input species. A total of 66 samples and 13 species were included in PMF analysis, which was an adequate sample size to obtain a statistically reliable PMF result. Details of PMF parameter settings are provided in the Supplementary Material.

**3 Results and discussion**

**3.1 Total-HULISWS mass concentration and HULISWS-associated the DTT activity of HULISWS**

In this study, the concentrations of total HULISWS mass concentration and HULISWS associated DTT activity of HULISWS in 66 PM2.5 samples were quantified. The annual average concentration of total HULISWS in Beijing measured in this study was 5.66  $\mu$ g m-3 (median: 4.30, range: 1.08–22.36  $\mu$ g m-3). This was approximately 20% higher than those measured in three other Chinese cities: 4.83  $\mu$ g m-3 in Guangzhou (Kuang et al., 2015), 4.71  $\mu$ g m-3 in Nansha (Kuang et al., 2015), and 4.69  $\mu$ g m-3 in Lanzhou (Tan et al., 2016). A clear temporal variation of total-HULISWS mass concentration was observed (Figures 1, 2), with significantly higher levels (*p* < 0.05, Mann–Whitney test) in the heating season (November through March; average 7.93, median 6.15  $\mu$ g m-3) than in the non-heating season (April through October; average 3.72, median 2.86  $\mu$ g m-3). This could be mostly attributed to the intensive coal and biomass burning activities performed for residential heating during the heating season. In addition, the lower temperatures and mixing heights during the heating season could also favorfavour the formation of particle-bound HULISWS species. However, the contributions of total HULISWS to organic matter (OM, calculated by <del>OC</del>-multiplying OC with the ratio of 1.98 and 1.50 for the heating and non-heating seasons, respectively, Xing et al., 2013) in PM2.5 are slightly lower during the heating season (21.8% ± 13.5%) than that during the non-heating season (27.4% ± 12.0%, Figure 1), indicating higher levels of -other-combustion-generated organic compounds other than HULISWS were emitted in the heating seasons <del>other than HULISWS aveil</del>.

For HULISws-associated-The extrinsic DTT activity of HULISws, they exhibited similar temporal variation as HULISws (Figure 2), with significantly higher levels in the heating season (average 0.073, median 0.063 nmol min-1 m-3) than in the non-heating season (average 0.031, median 0.029 nmol min-1 m-3). Because most of the inorganic ions were not retained by the HLB cartridge and the remaining metals in the HULISws effluent were chelated by DTPA, the DTT activity measured here could be attributed entirely to the DTT active moieties in HULISws. To-further investigate the intrinsic ROS generation abilityDTT activity of HULISws, the DTT consumption rate was normalized\_describes the intrinsic ROS-generation ability of HULISws, for HULISws mass (DTTm expressed in units of pmol min-1 per µg HULISws (median 9.02, range 2.74–25.8 pmol min-1 per µg HULISws), which was higher than the reported average DTTm activity (6.4 ± 1.2 pmol min-1 per µg HULISws) in for six PM2.5 
[revised manuscript text omitted]
 NOX, O3, SO42-, particle acidity (Hp+), and particle-phase liquid water content (LWCp) on the-HULISWS resolved in the vehicle emissions factor (HULISWS\_VE; the calculation of Hp+ and LWCp, and the MLR analysis results are provided in the Supplementary Material). NOX was found as the only statistically significant factor that was positively correlated to HULISWS\_VE with a regression coefficient of 0.012 (p < 0.001; Table S22 in the Supplementary Material), suggesting that a 1 µg m-3 increase in NOX was associated with a 0.012 µg m-3 increase in HULISWS\_VE, when holding other covariates unchanged. In fact, vehicle exhaust was the major source of ground level NOX (>60%) in Beijing, even in the heating season (Lin et al., 2011). A higher level of NOX was observed during the heating season than during the non-heating season due to a lower boundary layer and weaker vertical mixing (Figure S6 in the Supplementary Material). Kautzman et al. (2010) found that ring-opening oxygenated products with one benzyl group, which could be retained by the HLB cartridge and were considered as HULISWS components, were predominantly formed from the photo-oxidation of PAHs under high NOX conditions. Thus, the higher levels of NOX in the heating season led to higher levels of secondarily produced HULISWS\_VE, indicating a synergistic effect of primary emission and the secondary aging process from vehicle exhaust. Furthermore, the presence of 4M5NC and 3M6NC, SOA markers of cresol, in this factor confirmed that a certain fraction of HULISWS\_VE was secondarily formed.

In addition to the four combustion-related sources, one secondary source was apportioned by PMF, contributing 30.1% of HULISWS throughout the year. MLR analysis was conducted to evaluate the effects of  $O_3$ ,  $NO_X$ ,  $SO_4^{2^-}$ ,  $H_p^+$ , and LWCp on the secondary formation of HULISWS (HULISWS-SEC). Sulfate was found to be the most significant factor with a regression coefficient of 0.066 (Table S43 in the Supplementary Material). This may be due to the predominant role of sulfate in the particle-phase formation of organosulfates, one important HULISWS component (Xu et al., 2015), through both nucleophilic addition reactions and the salting-in effect (Lin et al., 2012; Riva et al., 2015). Results from the MLR analysis also indicated that an increase of 1  $\mu$ g m-3 O3 led to an increase of 0.028  $\mu$ g m-3 HULISWS\_SEC. Gaseous highly oxidized multifunctional organic compounds (HOMs) were characterized in the ozonolysis of  $\alpha$ -pinene in smog chamber experiments (Zhang et al., 2015). It was suggested that, after partitioning to the particle phase, these HOMs could undergo rapid accretion reactions to form oligomers containing multiple carboxylic acid and ester groups, which served as good HULISWS\_SEC was produced in the higher concentrations of O3 in the non-heating season (Figure S7 in the Supplementary Material), together with higher biogenic VOCs emissions and temperature as well as more intense solar radiation, a larger amount of HULISWS\_SEC was produced in the non-heating season (2.01  $\mu$ g m-3) than in the heating season (1.41  $\mu$ g m-3).

**3.5 Source-specific contributions to DTT activity of HULISWS**

To gain quantitative insights into the potential health impacts of different  $HULIS_{WS}$  sources, source-specific contributions to  $HULIS_{WS}$ -associatedextrinsic DTT activity of  $HULIS_{WS}$  were assessed using PMF-result. The strong correlation ( $R^2 = 0.78$ ; Figure S4 in the Supplementary Material) between measured and predicted DTT activity suggested reliable predictions.

As shown in Figure 4B, Similar to the source apportionment results of HULISWS, biomass burning was identified as the major contributor to HULISWS associated DTT activity in the heating season, and secondary formation was the most important source in the non heating season (Figure 4B). the four combustion-related sources accounted for 75% of HULISWS associatedthe extrinsic DTTredox activity of HULISWS throughout the year, of which biomass burning contributed 33.6%, followed by vehicle emissions (18.5%), waste incineration (18.5%), and coal combustion (4.1%). The extrinsic DTT activity of HULISWS describes the redox activity of HULISWS on air volume basis (E.q.(1)), which is reflective of human exposure to HULISWS; while the intrinsic DTT activities of HULISWS is on mass basis and is more important for assessing the intrinsic toxicity HULISWS from various sources. Furthermore, tThe intrinsic DTT activities of the HULISWS from the five identified sources were derived calculated (E.q.(2)). HULISWS from vehicle emissions constituted was found to be the most DTTROS-active HULISWSs with a maximum activity of (12.0 pmol min-1 per µg HULISWS VE), followed by waste incineration (9.25 pmol min-1 per µg HULISWS WI, biomass burning (9.10 pmol min-1 per  $\mu$ g HULISWS BB), secondary formation (7.45 pmol min-1 per  $\mu$ g HULISWS SEC), and coal combustion (6.22 pmol min-1 per  $\mu$ g HULISWS SEC).

Similar to the source apportionment results of HULISWS, biomass burning was identified as the major-leading contributor to extrinsic HULISWS associated DTT activity of HULISWS in the heating season (39.4%, 0.015 nmol min-1 m-3), and throughout the yearsecondary formation was the most important source in the non-heating season (33.6%, 0.017 nmol min-1 m-3Figure 4B). During biomass burning, highly oxidized organic compounds with quinone, hydroxyl, and carboxyl groups were directly produced (Fan et al., 2016). Moreover, some of the VOCs emitted from biomass burning could undergo further reactions and generate highly redox-active products, for example, hydroxyquinones formed through •OH radical oxidation (McWhinney et al., 2013). Those compounds, such as quinones and hydroxyquinones, which could be extracted into the HULISWS fraction and lead to DTT consumption (Chung et al., 2006; Verma et al., 2015a). AdditionallyMoreover, Wang et al. (2017) found large amounts of nitrogen-containing organic compounds (NOCs) including nitroaromatics and nitrogen-containing bases in HULISWS from biomass burning. The nitrite group next to aromatic ring in the nitrogen-containing alkaloids bases emitted from biomass burning could also enhance the ROS-generation ability of HULISWS BB.

Secondary formation was the most important source for the extrinsic DTT activity of HULISWS in the non-heating season  $(44.1\%, 0.015 \text{ nmol min}^{-1} \text{ m}^{-3})$  and the second largest contributor throughout the year  $(25.3\%, 0.013 \text{ nmol min}^{-1} \text{ m}^{-3})$ . A few smog chamber experiments have been carried out to investigate the ROS activity of SOA from various hydrocarbon precursors, and the intrinsic DTT activity values of several biogenic SOA systems (i.e. isoprene,  $\alpha$ -pinene, and  $\beta$ -caryophyllene) were found to be within the range of 2 to 30 pmol min-1 per µg SOA (Kramer et al., 2016; Tuet et al., 2017). Tuet et al. (2017) also observed a much higher intrinsic DTT activity of naphthalene SOA than that of biogenic SOA, and suggested that this was probably due to the aromatic species, especially nitroaromatics, in naphthalene SOA. The intrinsic DTT activity of HULISWS SEC measured in this study is 7.45 pmol min-1 per  $\mu$ g HULISWS SEC, which is within the reported intrinsic DTT activity range of biogenic SOA. Moreover, results from MLR analysis indicated that both sulfate and ozone were positively correlated with HULISWS SEC (Table S4), suggesting that HULISWS resolved in this factor could mainly consist of some less ROS-active SOA components, such as organosulfates (Chen et al., 2011; Lin et al., 2012). Although chamber experiments reported the formation of ROS-active HOMs or organic peroxides through the ozonolysis of biogenic VOCs (Docherty et al., 2005; Zhang et al., 2015), the production yields of these peroxides were generally low and thus could not have a major influence on the DTT activity of HULISWS SEC. However, since secondary formation predominated in HULISws formation (Figure 4A), especially in the non-heating season (50.1%), even with a lower intrinsic DTT activity, secondary aerosol formation still serves as a significant contributor to HULISWS-associated redox activity in Beijing. It should be noted that the contributions of secondary formation processes to both HULISWS and DTT activity of HULISWS could even have been underestimated in this study, because HULISWS secondarily formed through the aging of biomass burning and vehicle emissions was resolved in factor B and D and could not be accurately quantified.

Although vehicle emission just contributed 18% to extrinsic DTT activity of HULISWS throughout the year (18.5%, 0.009 nmol min-1 m-3), HULISWS VE has the highest intrinsic DTT activity among all sources (12.0 pmol min-1 per  $\mu$ g HULISWS VE). -<del>To</del> further investigate the intrinsic ROS-generation ability of HULISWS, the DTT consumption rate was normalized for HULISWS mass (DTTm, expressed in units of pmol min-1 per  $\mu$ g HULISWS (Verma et al. 2014). The average intrinsic DTT activity of HULISWS in Beijing was 9.91 pmol min-4 per  $\mu$ g HULISWS (median 9.02, range 2.74–25.8 pmol min-4 per  $\mu$ g HULISWS), which was higher than the reported average DTTm activity (6.4 ± 1.2 pmol min-4 per  $\mu$ g HULISWS) of six PM2.5-samples collected

during winter in Guangdong, China (Dou et al., 2015). This difference might be attributed to the different chemical components and sources of HULISws-in these two regions.

Furthermore, the intrinsic DTT activities of the HULISWS from the five sources were derived. HULISWS from vehicle emissions constituted the most ROS-active HULISWS, with a maximum activity of 12.0 pmol min-4-per  $\mu$ g HULISWS\_VD followed by waste incineration (9.25 pmol min-4-per  $\mu$ g HULISWS\_WA), biomass burning (9.10 pmol min-4 per  $\mu$ g HULISWS\_DD), secondary formation (7.45 pmol min-4-per  $\mu$ g HULISWS\_SEC), and coal combustion (6.22 pmol min-4 per  $\mu$ g HULISWS\_CC). Similarly, Bates et al. (2015) revealed that the water-soluble PM2.5 (WS PM2.5) from gasoline vehicle emissions had the highest intrinsic DTT activity, probably due to the oxygenated OC and metals on gasoline particles. Verma et al. (2009) also observed a higher aerosol oxidative potential from the aged particles of traffic exhaust than those directly emitted, and a strong correlation was observed between oxygenated organic acids and vehicle-related redox activity. As shown in Figure 2D, most of the two methyl nitrocatechol markers were resolved in the vehicle emissions factor and HULISWS\_VE was found to be significantly correlated with NOX, therefore the high intrinsic ROS activity of HULISWS\_VE is believed to be mostly due to the highly oxygenated OC content, especially the highly redox-active nitroaromatics (Tuet et al., 2017). In the present study, vehicle emission was found to be the highest redox active source for HULISWS\_VE a large fraction of WS PM2.5. However, because the remaining water soluble metals in HULISWS-were chelated through DPTA, the high intrinsic ROS activity of HULISWS\_VE is believed to be mostly due to the highly oxygenated OC content in HULISWS\_VE-

Waste incineration was also\_another important primary source of the extrinsic\_HULISws-related-DTT activityof HULISws (20.5% in the non-heating season and 17.4% in the heating season), and its intrinsic HULISws ROS activity was-even slightly higher than that from biomass burning. Mohr et al. (2009) examined the elemental ratio of aerosols emitted from different sources. They found that particles from plastic burning had a higher O/C ratio (0.08) than those from diesel (0.03) and gasoline (0.04) combustion, indicating a more oxidized feature of aerosols emitted through refuse burning (Mohr et al., 2009). Considering that incineration will play an increasingly important role in waste treatment in Beijing in the following years (National Development and Reform Comission, 2016), concern should be directed to the potential threat of trash burning to public health.

Although HULISWS\_SEC was less DTT active than HULISWS\_VE, HULISWS\_WI, or HULISWS\_BB, secondary aerosol formation served as the second largest contributor (25.3%) to HULISWS associated redox activity throughout the year. Higher levels of DTT activity mediated by HULISWS\_SEC were observed in the non heating season (0.015 nmol min=1-m=3) than in the heating season (0.011 nmol min=1-m=3), accounting for 44.1% and 14.5% of HULISWS DTT activity in each season, respectively. The relatively low intrinsic DTT activity of HULISWS\_SEC may be mostly attributed to its abundance of biogenic SOA components such as organosulfates and organonitrates (Chen et al., 2011), which were found to have negligible ROS generating ability (Kramer et al., 2016). Although chamber experiments reported the formation of ROS active HOMs or organic peroxides through the ozonolysis of biogenic VOCs (Docherty et al., 2005; Zhang et al., 2015), the production yields of these peroxides were generally low and thus could not have a major influence on the DTT activity of HULISWS\_SEC.

In summary, four combustion-related sources and one secondary formation source of  $PM_{2.5}$ -bound HULISWS and their associated ROS potential activity were identified by PMF-in this study. Biomass burning (32.7%) and secondary aerosol formation (30.1%) were the major contributors to HULISWS in Beijing. For the first time, waste incineration was identified as an important source of HULISWS, with a considerable and stable contribution to HULISWS throughout the year (17.7%). Regarding ROS-generation

potential,  $HULIS_{WS}$  from vehicle emissions was identified as the most ROS-active, and  $HULIS_{WS}$  from secondary aerosol formation showed a lower intrinsic DTT ability than those of most primary sources except for coal combustion. Such variations in the ROS-generation ability of  $HULIS_{WS}$  from different sources will be relevant for future inquiries into more detailed chemical speciation of  $HULIS_{WS}$ , their roles in ROS generation, and the possible oxidation mechanisms involved.

**Supplementary Material.** Information on chemical analysis; PMF source apportionment; MLR analysis together with Table S1-S43 and Figure S1-S7 are provided.

Acknowledgement. This work was supported by the National Natural Science Foundation of China (NSFC-(21477102, 21322705 and 41421064), the Joint NSFC-ISF Research Program (41561144007), the Ggeneral Rresearch Fund of Hong Kong Research Grant Council (12304215, 12300914 and 201212), the Ministry of Science and Technology of China Grants (973 program; 2015CB553401), the Faculty Research Grant from Hong Kong Baptist University (FRG2/16-17/041), and Research and Development of Science and Technology in Shenzhen (JCYJ 20140419130357038 and JCYJ 20150625142543472). The author would like to thank Binyu Kuang from Hong Kong University of Science and Technology for HULISWS quantification.

[revised manuscript text omitted]

---

## Author Response (AR3)

**Point-by-point Response to Co-Editor Comments on**

**"Sources and oxidative potential of water-soluble humic-like substances (HULIS$_{WS}$) in fine particulate matter (PM$_{2.5}$) in Beijing"**

**Co-Editor Comments to the Author:**

Reviewer 3 made the comment: 3. Secondly, unfortunately the authors apparently quantified DTT activities using a non-standard method, ie that of Li et al., 2009, which differs from the original DTT protocol described by Cho et al. (2005). The Cho method is widely utilized and is the basis for most DTT activities reported in the published literature. This means that the DTT activities reported here cannot necessarily be directly compared to generally published results, unless some type of conversion factor is given. This should be noted in the paper, discussed in detail (ie, difference in protocols explained), and optimally, a conversion factor given based on further experimental work by the authors.

You then gave a detailed answer to this. Can you please reproduce this in an appropriate manner please in the revised, MS, as you didn't add any text to the MS in response to this important point raised by the referee.

**Author Response:** Thanks for the comments. In the revised manuscript, we have added the following statement: "Considering the reaction between DTNB and DTT was very fast, the absorption could reach its maximum value immediately and stay stable for more than 2 hours (Li et al., 2009). So we followed the same protocol described in Li et al. (2009) with the elimination of quenching step described in Cho et al.'s method (2005), and conduct measurement at 412 nm within 30 min using an ultraviolet-visible (UV-Vis) spectrophotometer (8453, Hewlett Pakard, Palo Alto, CA, USA)" (lines 110-114).

For points 12 and 13, you simply gave the response: Author Response: Please refer to the above response to comment 4.

I would like you please to explicitly respond to each of these as separate responses, and if there is text in the response to comment 4 that is relevant, please cut and paste the relevant section. It is important that all comments have an explicit response, as 12 and 13 are different questions to comment 4

**Author Response:** Thanks for your advice. We now respond to points 12 and 13 separately.

**Point 12.** The discussion of possible reasons why DTT activities of the secondary factor are so high in summer needs more attention. Are the authors saying that biogenic SOA is the cause? There are papers that make this dubious claim (Kramer, A., W. Rattanavaraha, Z. Zhang, A. Gold, J. D. Surratt, and Y.-H. Lin (2016), Assessing the oxidative potential of isoprene-derived epoxides and secondary organic aerosol, Atmos. Env., 130, 211- 218.).

**Author Response:** Actually our result on the intrinsic DTT activity (DTT consumption normalized by SOA mass) of HULIS$_{WS-SEC}$ is consistent with findings from previous papers on the ROS activity of biogenic SOA.

In the paper mentioned by the reviewer, Kramer et al. (2016) evaluated the intrinsic DTT activity of SOA from isoprene and methacrolein. Later, Tuet et al. (2017) also investigated intrinsic DTT activity values of several biogenic SOA systems, i.e. isoprene, α-pinene, and β-caryophyllene. Based on their findings, the intrinsic DTT activity values of biogenic SOA were reported to be in the range of 2 to 30 pmol min$^{-1}$ per μg SOA, depending on the individual biogenic VOC precursors. The intrinsic DTT activity of HULIS$_{WS\_SEC}$ measured in this study is 7.45 pmol min$^{-1}$ per μg HULIS$_{WS\_SEC}$, which is within the reported intrinsic DTT activity range of biogenic SOA. Multilinear regression (MLR) analysis has been conducted to assess the effects of NO$_X$, O$_3$, SO$_4^{2-}$, particle acidity (H$_p^+$), and particle-phase liquid water content (LWC$_p$) on HULIS$_{WS\_SEC}$, and sulfate was found to be the most significant factor with a regression coefficient of 0.066. This suggests that HULIS$_{WS}$ resolved in this factor could mainly consist of some less ROS-active SOA components, such as organosulfates (Chen et al., 2011; Lin et al., 2012).

Although the intrinsic DTT activity of HULIS$_{WS\_SEC}$ is lower than most of the combustion-related sources identified in this study, secondary formation predominated in HULIS$_{WS}$ formation (Figure 4A), especially in the non-heating season (50.1%). Therefore, the contribution of secondary aerosol formation to extrinsic DTT activity, which is the product of intrinsic DTT activity and HULIS$_{WS\_SEC}$ mass concentration, is significant in Beijing, especially in the non-heating season.

In the revised manuscript, lines 296-313, we added: "Secondary formation was the most important source for the extrinsic DTT activity of HULIS$_{WS}$ in the non-heating season (44.1%, 0.015 nmol min$^{-1}$ m$^{-3}$) and the second largest contributor throughout the year (25.3%, 0.013 nmol min$^{-1}$ m$^{-3}$). A few smog chamber experiments have been carried out to investigate the ROS activity of SOA from various hydrocarbon precursors, and the intrinsic DTT activity values of several biogenic SOA systems (i.e. isoprene, α-pinene, and β-caryophyllene) were found to be within the range of 2 to 30 pmol min$^{-1}$ per μg SOA (Kramer et al., 2016; Tuet et al., 2017). Tuet et al. (2017) also observed a much higher intrinsic DTT activity of naphthalene SOA than that of biogenic SOA, and suggested that this was probably due to the aromatic species, especially nitroaromatics, in naphthalene SOA. The intrinsic DTT activity of HULIS$_{WS\_SEC}$ measured in this study is 7.45 pmol min$^{-1}$ per μg HULIS$_{WS\_SEC}$, which is within the reported intrinsic DTT activity range of biogenic SOA. Moreover, results from MLR analysis indicated that both sulfate and ozone were positively correlated with HULIS$_{WS\_SEC}$ (Table S4), suggesting that HULIS$_{WS}$ resolved in this factor could mainly consist of some less ROS-active SOA components, such as organosulfates (Chen et al., 2011; Lin et al., 2012). Although chamber experiments reported the formation of ROS-active HOMs or organic peroxides through the ozonolysis of biogenic VOCs (Docherty et al., 2005; Zhang et al., 2015), the production yields of these peroxides were generally low and thus could not have a major influence on the DTT activity of HULIS$_{WS\_SEC}$. However, since secondary formation predominated in HULIS$_{WS}$ formation (Figure 4A), especially in the non-heating season (50.1%), even with a lower intrinsic DTT activity, secondary aerosol formation still serves as a significant contributor to HULIS$_{WS}$-associated redox activity in Beijing. It should be noted that the contributions of secondary formation processes to both HULIS$_{WS}$ and DTT activity of HULIS$_{WS}$ could even have been underestimated in this study, because HULIS$_{WS}$ secondarily formed through the aging of biomass burning and vehicle emissions was resolved in factor B and D and could not be accurately quantified."

**Point 13.** Given that most sulfate is in this factor (and surprisingly in contrast to the coal combustion factor), it seems that the factor is really all about oxidation processes (as the name for the factor implies). One explanation is that this factor really has contributions from all source, such as coal, vehicles, and even biomass burning, given the loss of levoglucosan through oxidation (although this source is lower in summer). Verma et al, 2015a noted the strong dependence of aging on enhanced quinone DTT activity. This factor may just represent this process. Despite significant discussion of the other factors, this factor is not considered sufficiently give the large role it plays in summer DTT activities, a major finding of the paper.

**Author Response:** The high contribution of secondary aerosol formation to extrinsic DTT activity is mainly due to its significant contribution to $HULIS_{WS}$ mass, especially in summer (50.1%). To investigate the possible formation mechanisms of $HULIS_{WS\_SEC}$, MLR analysis was conducted to evaluate the influences of $O_3$, $NO_X$, $SO_4^{2-}$, $H_p^+$, and $LWC_p$ on $HULIS_{WS\_SEC}$. Both sulfate and ozone were found to be statistically significant factors that positively correlated with $HULIS_{WS\_SEC}$, which suggested the secondary formation of both organosulfates and highly oxygenated organic compounds. In the revised manuscript, we've added: "In addition to the four combustion-related sources, one secondary source was apportioned by PMF, contributing 30.1% of $HULIS_{WS}$ throughout the year. MLR analysis was conducted to evaluate the effects of $O_3$, $NO_X$, $SO_4^{2-}$, $H_p^+$, and $LWC_p$ on the secondary formation of $HULIS_{WS}$ ($HULIS_{WS-SEC}$). Sulfate was found to be the most significant factor with a regression coefficient of 0.066 (Table S4 in the Supplementary Material). This may be due to the predominant role of sulfate in the particle-phase formation of organosulfates, one important $HULIS_{WS}$ component (Xu et al., 2015), through both nucleophilic addition reactions and the salting-in effect (Lin et al., 2012; Riva et al., 2015). Results from the MLR analysis also indicated that an increase of 1 μg m$^{-3}$ $O_3$ led to an increase of 0.028 μg m$^{-3}$ $HULIS_{WS\_SEC}$. Gaseous highly oxidized multifunctional organic compounds (HOMs) were characterized in the ozonolysis of α-pinene in smog chamber experiments (Zhang et al., 2015). It was suggested that, after partitioning to the particle phase, these HOMs could undergo rapid accretion reactions to form oligomers containing multiple carboxylic acid and ester groups, which served as good $HULIS_{WS}$ candidates. Considering the higher concentrations of $O_3$ in the non-heating season (Figure S7 in the Supplementary Material), together with higher biogenic VOCs emissions and temperature as well as more intense solar radiation, a larger amount of $HULIS_{WS\_SEC}$ was produced in the non-heating season (2.01 μg m$^{-3}$) than in the heating season (1.41 μg m$^{-3}$)" (lines 259-271).

Reference:

Chen, X., Hopke, P. K. and Carter, W. P. L.: Secondary organic aerosol from ozonolysis of biogenic volatile organic compounds : Chamber studies of particle and reactive oxygen species formation, Environ. Sci. Technol., 45(1), 276–282, 2011.

Cho, A. K., Sioutas, C., Miguel, A. H., Kumagai, Y., Schmitz, D. A., Singh, M., Eiguren-Fernandez, A. and Froines, J. R.: Redox activity of airborne particulate matter at different sites in the Los Angeles Basin, Environ. Res., 99(1), 40–47, 2005.

[revised manuscript text omitted]

**Reviewer #2:**

It's well known that the toxicity of PM$_{2.5}$ is greatly related to its chemical composition and pollution sources. This work analyzed PM$_{2.5}$ samples collected in Beijing during a one-year period, and the levels and oxidative potential (indicated by DTT) of a major water-soluble PM$_{2.5}$ component, i.e. water-soluble humic-like substances (HULIS$_{WS}$), were reported. With the aid of various characteristic source tracers, PMF was applied to apportion the major sources of both HULIS$_{WS}$ and its associated DTT activity. There are some new and interesting findings. The major sources of both HULIS$_{WS}$ and DTT activity were coal combustion, biomass burning, traffic exhaust, waste incineration, and secondary formation. Waste incineration was probably identified as a contributor to HULIS$_{WS}$ for the first time. Moreover, HULIS$_{WS}$ from vehicle emissions was found as the most ROS-active, and HULIS$_{WS}$ from secondary aerosol formation showed a lower intrinsic DTT ability than those of most primary sources except for coal combustion. This paper is well-written. The study is clear, informative, and novel in general, and the major data and their interpretation are scientifically sound. I suggest it to be considered by ACP for publication if the following concerns could be addressed.

**Author Response:** Thanks for the comments.

1. Line 22: Is it necessary to define the waste as "plastic waste", as terephthalic acid is a marker of plastics?

**Author Response:** Since terephthalic acid is an important industrial material for making PET (polyethylene terephthalate) plastics, they have been found to be abundant in plastic burning smokes. In developing countries, plastic materials are dumped as domestic waste, which makes the waste plastic-enriched. In this case, terephthalic acid serves as a marker of plastic burning and plastic-enriched domestic waste burning. Therefore, we will keep the name of this factor as "waste incineration".

2. Line 39-40: Add "an" before electron and "a" before continuous.

**Author Response:** This revision has been done.

In the revised manuscript, line 40-41 (in the manuscript file of "BJ HULIS_clear"; the same below): "The reversible redox sites in HULIS$_{WS}$ fraction could serve as an electron transfer intermediate and lead to continuous production of ROS (Lin and Yu, 2011)."

3. Line 71: If the samples were taken every 6 days for a one-year period, there should be 60 samples. Why there were 66 samples as listed in line 123?

**Author Response:** In winter time, we collected several additional samples during severe polluted periods.

In the revised manuscript, line 70-71, we've added "…with several additional samples collected during severe polluted periods".

4. Line 100: How many individual hopanes have been identified? I guess the input species "hopane" in PMF must be the sum of all identified hopanes, right? What are the concentrations of hopanes? What are the water-soluble ions identified? The author may need to include a table in the supplementary information that provide levels of hopanes, water-soluble ions, EC and OC in the batch of $PM_{2.5}$ samples analyzed.

**Author Response:** Five individual hopanes were identified, and these 5 hopanes were lumped together and put into PMF analysis. The concentrations of the measured hopanes, ions, EC and OC were listed in Table S1 in the revised supplementary material.

In the revised manuscript, line 103-104, we've added: "Concentrations of hopanes, levoglucosan, water-soluble ions, EC and OC were listed in Table S1 in the Supplementary Material."

5. Line 108: DTPA was spiked to chelate transition metals. Could it also affect or even remove some $HULIS_{WS}$ components?

**Author Response:** In the previous work by Lin and Yu (2011), they assessed the DTT consumption by 1) $HULIS_{WS}$ sample without DTPA; 2) $HULIS_{WS}$ samples with DTPA; and 3) standard mixture solution of metals with concentrations similar to those measured in $HULIS_{WS}$ fraction. They found that the DTT consumption by residue metals in the absence of DTPA is similar to the difference between DTT consumption by equivalent amount of $HULIS_{WS}$ in the absence and presence of DTPA. Their result suggested that the DTPA spiked to chelate transition metals could barely affect the $HULIS_{WS}$ components. Although Dou et al. (2015) found that DTPA could suppress the DTT response of quinone by about 20%, considering this method is widely accepted in $HULIS_{WS}$ redox activity measurement, we adopt the same method in order to be consistent and make comparison with other $HULIS_{WS}$ studies (Dou et al., 2015; Lin and Yu, 2011). In the revised version, we mentioned the DTT activity of $HULIS_{WS}$ measured was underestimated and could not be directly comparable with that does not use DTPA as metal chelators.

In the revised manuscript, line 126-128, we've added: "Since DTPA was added to suppress DTT consumption by metals ions throughout the incubation process and may affect the DTT response of quinones (Dou et al., 2015), the DTT activity of HULIS$_{WS}$ measured here may be underestimated and is not directly comparable to those studies conducting DTT assay experiments without DTPA."

6. Lines 130 to 131: Were all the reference data observed during a one-year period and comparable to present study?

**Author Response:** Data reported for Guangzhou and Nansha sites were the average concentration of samples collected during a one-year period (2009 January – 2009 December). For data measured in Lanzhou, they were the average concentration of samples collected during two representative periods (June-July in summer and December in winter during 2012-2013).

[revised manuscript text omitted]